# New insights into combined surfzone, embayment, and estuarine bathing hazards

Christopher Stokes[1], Timothy Poate[1], Gerd Masselink[1], Tim Scott[1], Steve Instance[2]

[1]School of Biological and Marine Sciences, University of Plymouth, PL4 8AA, England
[2] Royal National Lifeboat Institution, West Quay Road, Poole, Dorset, BH15 1HZ, England

*Correspondence to*: Christopher Stokes (christopher.stokes@plymouth.ac.uk)

**Abstract.** Rip currents are the single largest cause of beach safety incidents globally, but where an estuary mouth intersects a beach, additional flows are created that can exceed the speed of a typical rip current, significantly increasing the hazard level for bathers. However, there is a paucity of observations of surfzone currents at estuary mouth beaches, and our understanding and ability to predict how the bathing hazard varies under different wave and tide conditions is therefore limited. Using field observations and process-based XBeach modelling at an embayed, estuary mouth beach, we demonstrate how surfzone currents can be driven by combinations of estuary discharge and wave-driven bathymetric and boundary rip currents under various combinations of wave and tide forcing. While previous studies have demonstrated the high hazard that rip currents pose, typically during lower stages of the tide, here we demonstrate that an estuary mouth beach can exhibit flows reaching 1.5 m/s – up to 50% stronger than typical rip current flows – with a high proportion (>60%) of simulated bathers exiting the surfzone during the upper half of the tidal cycle. The three-dimensional ebb shoal delta was found to strongly control surfzone currents by (1) providing a conduit for estuary flows that connect to headland boundary rips, and (2) acting as a nearshore bar system to generate wave-driven 'river-channel bathymetric rips'. Despite significant spatio-temporal variability in the position of the river channels on the beach face, it was possible to hindcast the timing and severity of past bathing incidents from model simulations, providing a means to forewarn bathers of hazardous flows.

## Introduction

*Estuary mouth beaches* are energetic environments where dynamic exchanges between marine and estuarine processes take place, resulting in complex hydrodynamics and a high degree of morphologic variability (Barnard and Warrick, 2010). For the present study we define them as wave-dominated beaches which feature an estuary that exits across the beach face, and distinguish them from *estuarine beaches*, which sit within enclosed estuary environments and where only fetch-limited, locally-generated waves are important (Nordstrom, 1992). Estuary mouth beaches are ubiquitous across the globe (Figure 1), with examples in meso- to macro-tidal environments in New Zealand (Hume and Herdendorf, 1988; Hume *et al.*, 2007) and the UK (Pye and Blott, 2014), as well as micro-tidal environments such as South Africa (Cooper, 2001) and Australia (Roy, 1984; Kench, 1999). Hume and Herdendorf (1988) class these environments as a *barrier (beach) enclosed estuary,* which are

typically small estuaries with low fluvial inflow, where the inlet is restricted by the barrier beach and direct exchange with the ocean only occurs near high tide (Hume and Herdendorf, 1988; Hume *et al.*, 2007). In the United Kingdom, 25% of designated bathing beaches feature a river or estuary. Of these 159 beaches, 29, including the present study site, are embayed with an alongshore distance between headlands of less than 3 km, concentrating the dynamic estuarine flows and sediment exchanges

over a relatively short length of coast.

Along the world's open coasts, rip currents have been identified as the largest cause of surfzone rescues and fatalities where incident records exist (Scott *et al.*, 2008; MacMahan *et al.*, 2011; Scott *et al.*, 2011; Brighton *et al.*, 2013), causing hundreds of drownings and tens of thousands of rescues globally each year (Castelle *et al.*, 2016). A rip current occurs when water set-up by wave breaking in the surfzone returns back out to sea in a concentrated and often fast-moving offshore flow (Brander,

1999; MacMahan *et al.*, 2006) and has the potential to carry water-users from the shallows out into deeper water. Previous research has demonstrated various forcing mechanisms for rip currents (Castelle *et al.*, 2016), including hydrodynamic instabilities in the surfzone ('shear instability rips' and 'flash rips'), bathymetric control of wave breaking and return flows ('channel rips' and 'focus rips'), and control of wave driven flows by headlands or other boundaries ('deflection rips', 'shadow rips', and 'cellular circulation'). When rips occur, a combination of factors, including circulation pattern, speed, and surfzone

retention influence the bathing hazard. Surfzone circulation typically varies between 'alongshore' flow at the shore which poses a low level of bathing hazard, 'rotational' flow where offshore and onshore currents circulate within the surfzone posing an intermediate level of bathing hazard, and 'exiting' behaviour where rip currents flow directly or obliquely seaward beyond the breaker zone to deeper water, representing the highest hazard to bathers (Scott *et al.*, 2014).

However, while studies across these rip types from various countries have observed rip velocities to average 0.4–1 m/s (Austin

*et al.*, 2010; MacMahan *et al.*, 2010; Castelle *et al.*, 2014; Scott *et al.*, 2014; McCarroll *et al.*, 2017; Moulton *et al.*, 2017a; McCarroll *et al.*, 2018; Mouragues *et al.*, 2020), flows within estuary channels on beaches have been observed to average 1–1.5 m/s and exceed 2 m/s during ebbing stages of the tide (Allen, 1971; Lessa and Masselink, 1995; Jiang *et al.*, 2013; Kastner *et al.*, 2019), with strong seaward flowing circulation. It is therefore surprising that, despite posing an equal or potentially even higher bathing hazard than rip currents, surfzone currents on estuary mouth beaches have received little attention in the

scientific literature. While several studies have investigated surfzone retention of river plumes in the context of larva, contaminant, and freshwater dispersal (Olabarrieta *et al.*, 2014; Rodriguez *et al.*, 2018; Kastner *et al.*, 2019), the effect of river or estuary discharges on surfzone bathing hazard remains unstudied.

Beach morphology classification schemes identify that intermediate beach states – those featuring three-dimensional nearshore bars and channels – are key locations for rip current activity and bathing hazard (Wright and Short, 1984; Lippmann and

Holman, 1990; Masselink and Short, 1993; Scott *et al.*, 2011). While beach classification schemes form the basis of our understanding of surfzone hazards and underpin lifeguard risk assessments in many nations (for example, the UK, Australia, and New Zealand), they ignore the presence of estuary mouths. Where an estuary mouth occurs on a beach, its channel and ebb-tidal delta form an integral part of the beach morphology (Hume *et al.*, 2007) and strongly influence the local flow velocities and nearshore circulation pattern, often promoting offshore flows during the ebb tidal phase (Cooper, 2001; Hume

*et al.*, 2007; Pye and Blott, 2014). A better understanding of nearshore circulation patterns and flow velocities at estuary mouth beaches is therefore required to fully understand combined surfzone and estuarine environments and the hazard they pose to bathers.

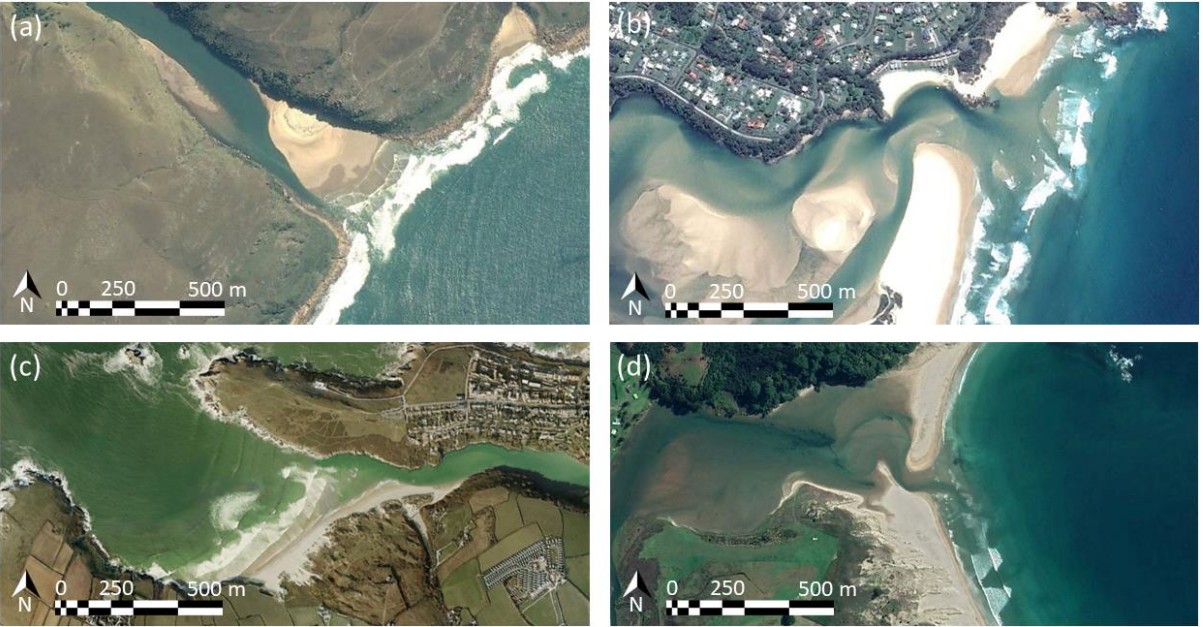

**Figure 1: Examples of estuary mouth beaches from around the world: (a) Mtentu, Eastern Cape, South Africa; (b) Tuross Head,**
**New South Wales, Australia; (c) Crantock, Cornwall, United Kingdom – the study site for this research; (d) Waikawau, Coromandel,**
**New Zealand. © Google Earth Pro.**

This contribution aims to investigate the interaction between estuarine flows and surfzone currents at an embayed, macrotidal, high-energy beach on the north coast of Cornwall in southwest England and evaluate the contribution of the various interacting processes on surfzone circulation and bathing hazard. A further aim is to develop a predictive system capable of forecasting
the level of bathing hazard to forewarn bathers prior to entering the beach. The study site is described in Section 2. In Section 3, a field experiment yielding Eulerian and Lagrangian flow measurements is described, leading to the development of a calibrated and validated hydrodynamic model of the estuary-mouth beach system. In Section 4, the model is used to explore surfzone circulation under various combinations of wave and tide forcing, and their influence on bathing hazard. In Section 5, a bathing hazards forecast system is described that represents the first operational forecast model for bathing hazards on an
estuary mouth beach. A discussion and conclusions are provided in Sections 6 and 7.

## 2 Study site

Crantock beach (Fig. 1c) is located in southwest England on the energetic and macrotidal north coast of Cornwall (Figure 2, upper panel). The beach is flanked on either side by East Pentire and West Pentire headlands, resulting in an embaymentisation ratio (alongshore length/headland length) of 0.4 (Masselink *et al.*, 2022). Because of the size of the headlands and angle of the

beach (310º) relative to the dominant wave approach (280º), the north end of the bay is exposed to more wave energy than the south, resulting in a strong gradient in wave height along the shore; a known precursor for headland boundary circulation (Castelle and Coco, 2012). Crantock experiences a mean spring tidal range of 6.3 m and is exposed to a mean and 1-year return period significant wave height $H_s$ of 1.5 m and 6.5 m, respectively.

Crantock is classed as an intermediate low-tide bar/rip beach following the classification scheme of Scott *et al.* (2011), but it features distinctly different low tide and high tide morphology. At low tide, the beach face is relatively planar, aside from pronounced boundary rip channels along each headland. At mid to high tide, the morphology is dominated by the ebb tide delta and the channel system associated with the Gannel Estuary. This results in complex three-dimensional features across the upper beach face, comprising often more than one river channel and multiple shoals (Figure 2, middle panel).

The Gannel Estuary has a catchment of 41 km² and flows onto Crantock beach at its northeast corner (Figure 2, upper panel) with mean and 5% exceedance riverine flow rates of 0.7 m³/s and 2 m³/s, as measured by a monitoring weir 2 km upstream of the tidal limit of the estuary (https://check-for-flooding.service.gov.uk/station/3135). Tidal discharges into and out of the estuary, however, are an order of magnitude larger than the riverine input (Section 2). For a number of decades, the river channel was artificially pinned against the northern headland by a small rock training wall (Figure 2, middle panel) to maintain a navigation channel across the beach for boats. However, following significant redistribution of the beach sediment during the unprecedented storms of the 2013/14 winter (Masselink *et al.*, 2016; Hird *et al.*, 2021), the river channel avulsed and now meanders laterally across the beach towards the south before discharging seaward through a channel that migrates between the south and north of the bay.

Royal National Lifeboat Institution (RNLI) lifeguards, who provide a lifeguard service at more than 250 UK beaches and have been present at Crantock Beach since 2001, have reported a rapid increase in both beach user numbers and lifeguard rescues in the years since the river avulsed. For example, lifeguard assistance and rescue numbers have increased from < 40 per year in 2014 to > 190 per year in 2018, including two fatalities when lifeguards were not present on the beach. The increase in rescues is likely to be driven in part by the increase in water users at the beach, but lifeguards also report that the river's new position has increased the level of bathing hazard. There is a particular concern that immediately before and after lifeguard patrol hours (10am–6 pm) estuary flows are at their strongest due to the coincidence of high spring tides at these times in the region. Bathers are therefore often exposed to strong ebb-tide flows (Figure 2, lower panel) without any lifeguard supervision. The new river course has carved deep troughs in the beach face which are submerged between mid and high tide, creating steep seabed gradients and spatially and temporally varying flows that are not visible to beach users.

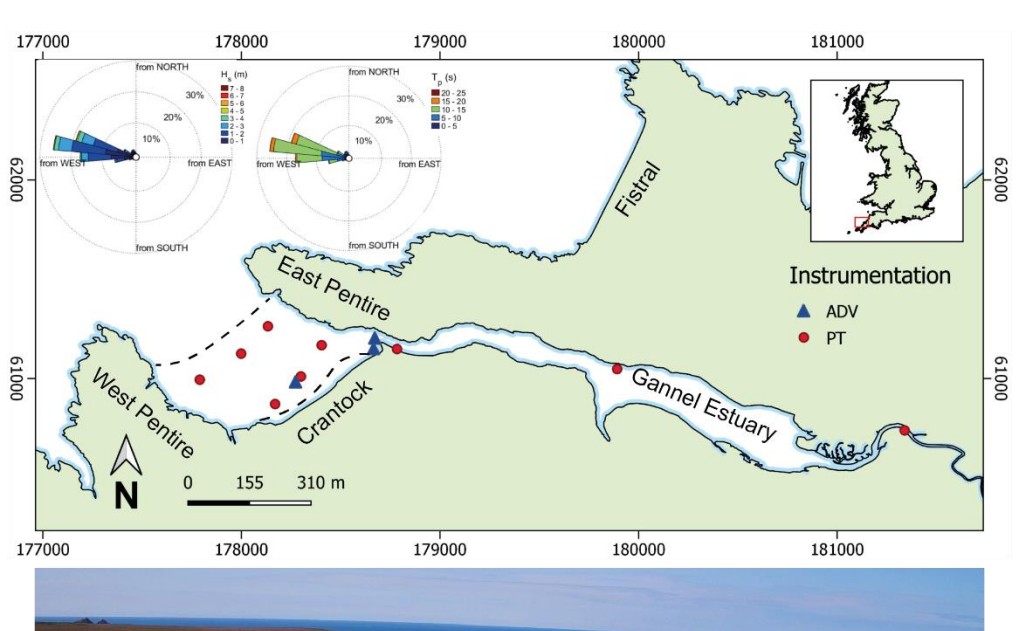

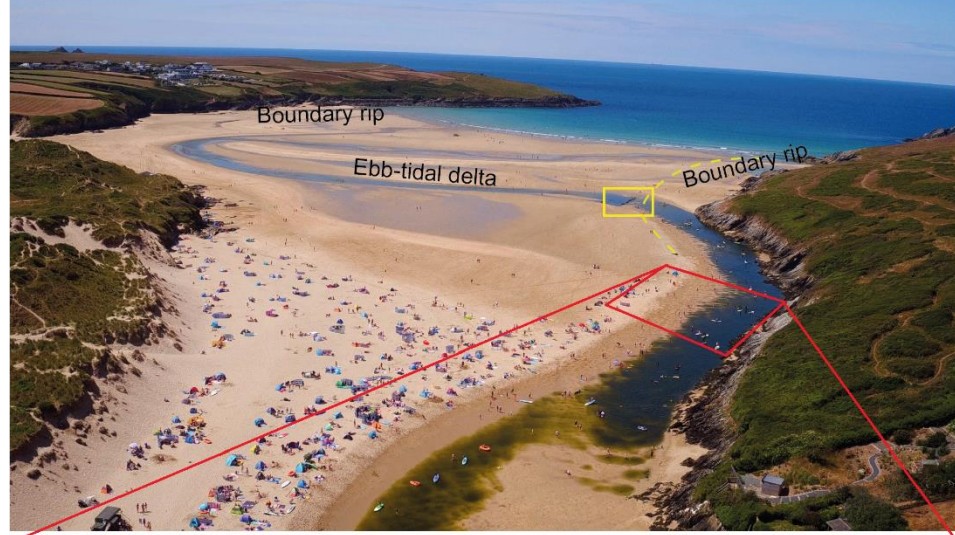

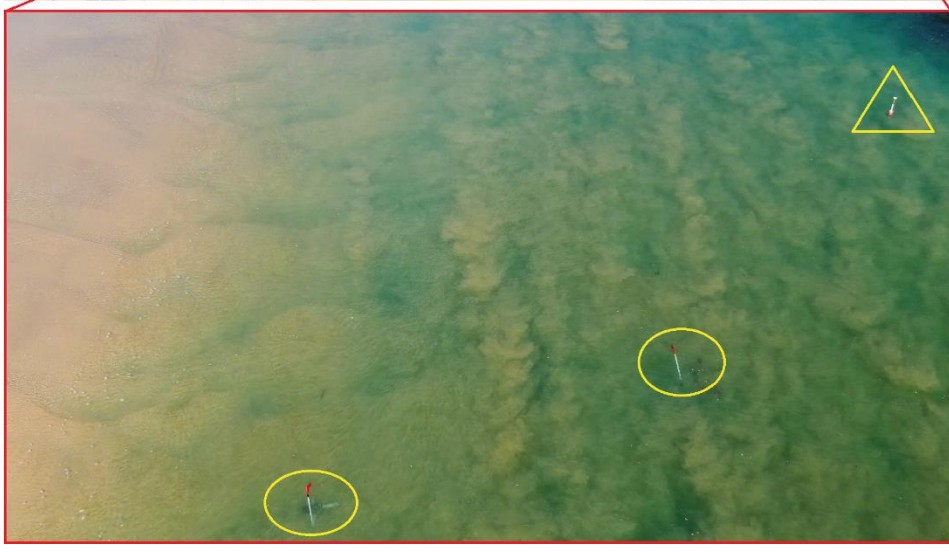

**Figure 2: (Upper panel) Crantock Beach and Gannel Estuary, field instrument locations, wave roses, and location in southwest England (OSGB36 eastings and northings). Mean low- and high-water lines are shown as dashed lines. (Middle panel) Aerial image showing the river channel across the intertidal beach, relic training wall (yellow rectangle), and former river channel position (yellow dashed line). (Lower panel) Aerial view of river flow during a high ebbing tide, with fixed ADV instrument rigs (yellow circles), and a Lagrangian GNSS drifter (yellow triangle). Note the high level of sediment suspension due to the strong flows.**

## 3 Methodology

### 3.1 Methodological approach

This study uses a combination of field data and numerical model simulations to investigate surfzone circulation patterns and bathing hazard. The field experiment allowed collection of both Eulerian and Lagrangian flow characteristics under average wave conditions combined with spring tides (Section 3.2) but was limited to the range of wave and tide conditions experienced over the three-day deployment. The numerical model provides the means to understand flow characteristics and bathing hazard under a much wider range of wave and tide conditions (Section 4.3.1), and with different realisations of the beach morphology (Section 4.3.2). Furthermore, the model provides the ability to 'switch off' the estuarine flows (Section 4.3.3) enabling us to disentangle the contribution of wave and estuary driven hydrodynamics on the surfzone circulation.

### 3.2 Field data

The field experiment (11$^{th}$ - 14$^{th}$ May 2021) focussed on the collection of topographic and bathymetric survey data, Eulerian wave and current measurements, and Lagrangian flow observations. Topographic data were collected using high-resolution aerial imagery captured with a DJI Phantom 4 RTK uncrewed aerial vehicle (UAV), equipped with accurate GNSS positioning system, flown across the site collecting multiple aerial images with an 80% overlap. The images were processed using photogrammetric techniques (structure-from-motion and multiview stereo) to create a digital elevation model (DEM) of the site. The DEM achieves a vertical RMSE of 0.03 m compared to independent spot checks against ground control points not used to geolocate the DEM during processing. The UAV flights were conducted around low water to maximise the coverage and visibility of the river channels. Once the data was captured, it was processed and translated onto a regular grid for further analysis.

For the full model domain to be mapped, a bathymetric survey was conducted at high tide to map the subtidal region, as well as overlap with the intertidal areas covered by the UAV. Multiple cross-shore transects ~25 m apart were recorded using a Valeport Midas Surveyor echosounder (acoustic frequency 210 kHz; sample rate 6 Hz), pole-mounted on an inflatable surf rescue vessel, with external Trimble RTK-GNSS positioning (Trimble 5800; sample rate 1 Hz). The bathymetric survey achieves a vertical RMSE of 0.1 m in the intertidal region, when compared to the previously mentioned ground control points. By merging the echosounder and UAV datasets (Section 3.4) the full survey region was extended down to ~10 m water depth, covering the full embayment (Fig. 1).

Eulerian measurements were collected by three Nortek Acoustic Doppler Velocimeters (ADVs) deployed on a rigid frame to allow current measurements to be logged ~0.1 m above the beach (Figure 2 and Figure 3), measuring alongshore, cross-shore and vertical flow velocities. Wave and tidal signals were logged using an array of nine pressure transducers (PTs) installed at bed level across the survey domain, three of which were co-located with ADVs (Fig. 2). All sensors were programmed to log at 4 Hz continuously. Outliers and spikes in the datasets were removed as part of quality control checks.

Lagrangian measurements were collected using GNSS-tracked surfzone drifters (Figure 3), which are designed to mimic a bather being carried by the surface flows (submerged approximately 0.5 m beneath the surface) and avoid surfing landward on waves. These were telemetered in real-time allowing shore based logging using QPS Qinsy software package (following Mouragues *et al.*, 2020). Six drifters were deployed at numerous locations multiple times across the survey area throughout the tidal cycle and were retrieved from the shallows before they ran aground. The raw data was then processed to remove time periods when the drifters were stationary on the beach, being deployed by hand or being recovered by the inshore survey boat. The cleaned data provides an x, y, t dataset where t = time.

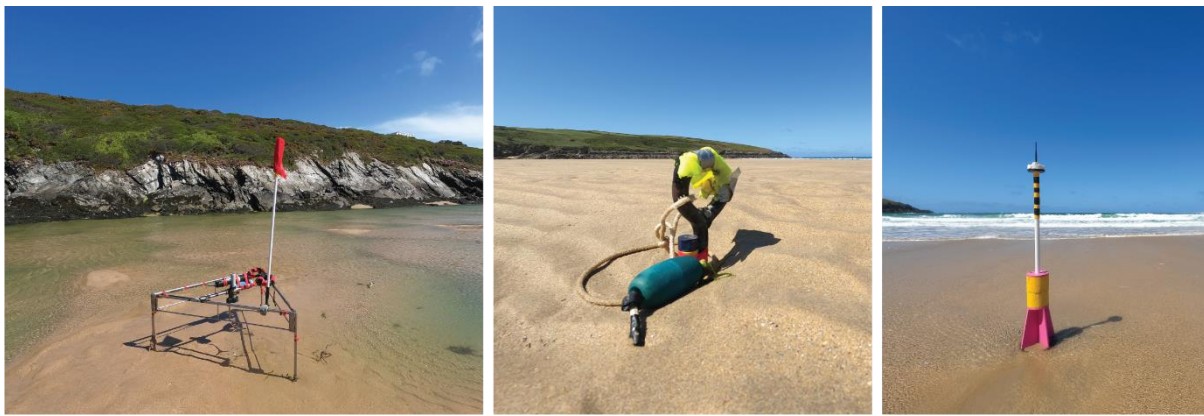

**Figure 3: Field instrumentation. (Left panel) Frame supporting the ADV logger, batteries, and sensor head ~0.1 m above the bed; (middle panel) pressure transducer sensor installed at bed level using a sand screw; and (right panel) a Lagrangian 'GNSS drifter' designed to map surface currents.**

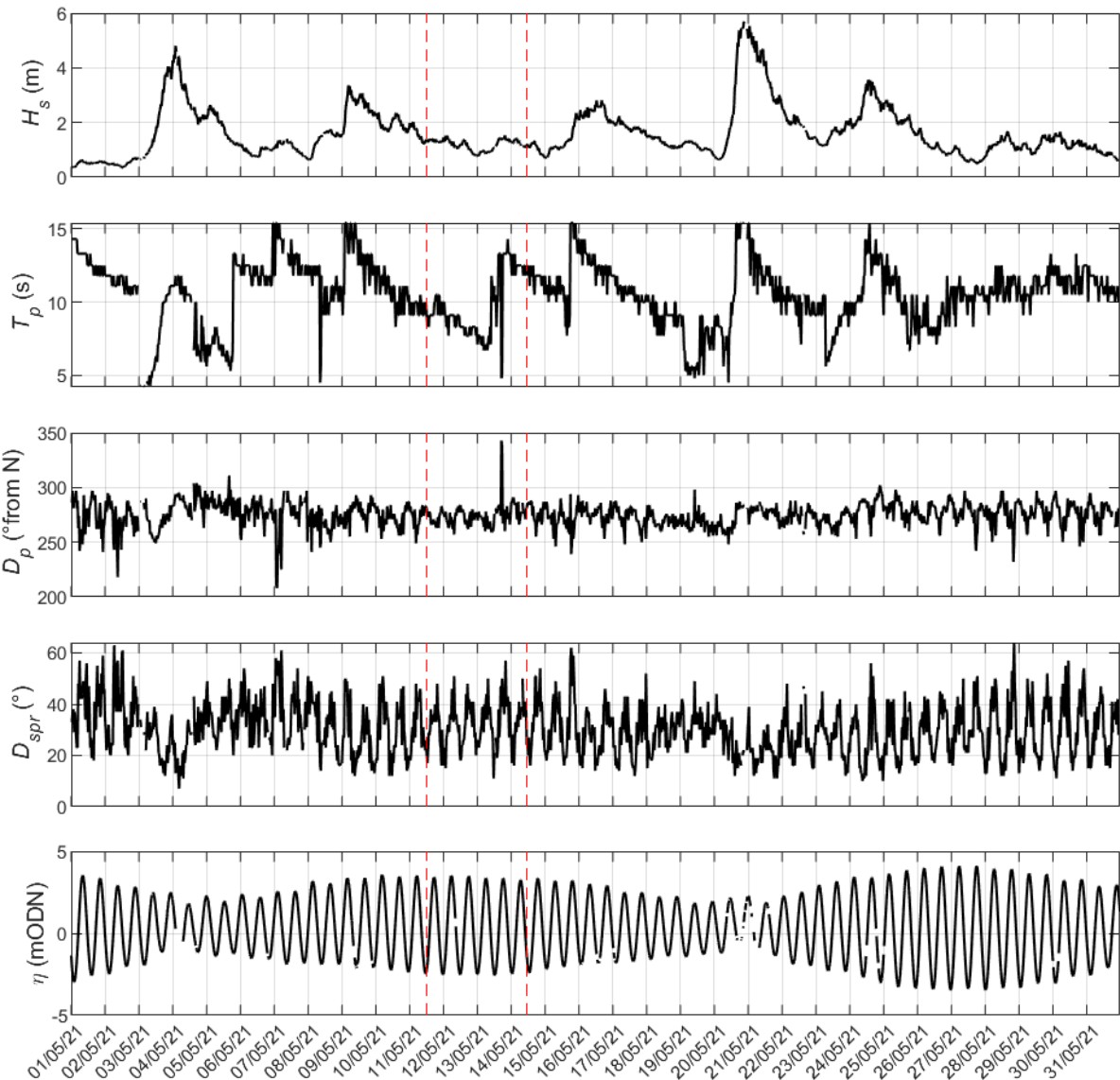

**Figure 4: Hydrodynamic conditions during the field deployment period (red dashed lines) measured at a waverider buoy 7 km south of Crantock and a tide gauge 29 km north of Crantock (https://coastalmonitoring.org/cco/). From top to bottom: significant wave height, $H_s$; peak wave period, $T_p$; direction of wave approach $D_p$; wave directional spread $D_{spr}$; and tidal water level, $\eta$.**


### 3.3 Numerical model

The process-based numerical model XBeach was used to simulate rip current and estuary-driven flows across Crantock beach. The model solves for the time-dependent short wave action-balance equations, roller energy equations, the non-linear shallow water equations of mass and momentum, sediment transport formulations, and morphological updating (Roelvink *et al.*, 2010).

Wave group dissipation is modelled (Roelvink, 1993; Daly *et al.*, 2012), and a roller model (Svendsen, 1984; Nairn *et al.*, 1991; Roelvink and Reniers, 2011) is used to represent the momentum carried after wave breaking. Radiation stress gradients (Longuet-Higgins and Stewart, 1962, 1964) then drive infragravity motion and unsteady currents in the model, which are solved with the non-linear shallow water equations (Phillips, 1977). In the 'surf beat' mode of operation used in this study, XBeach solves the variation of the short-wave envelope on the scale of individual wave groups (Roelvink *et al.*, 2018), which

has previously been found to reproduce measured hydrodynamics at dissipative and intermediate beaches favourably, including channel and boundary rip current behaviour (Austin *et al.*, 2013; McCarroll *et al.*, 2015; Scott *et al.*, 2016; Dudkowska *et al.*, 2020; Mouragues *et al.*, 2021). In the present study, morphological updating was switched off.

### 3.4 Model domain

A 2D-H (two horizontal dimensions, depth averaged) model was developed covering the full extent of Crantock beach (Figure

5), from the supratidal down to a seaward depth of 20 m below Ordnance Datum Newlyn (ODN). The domain was developed using the survey data described in Section 3.2 to cover the intertidal (-1 to +5 m ODN) and sub-tidal (-10 to +1 m ODN) regions. Repeat surveys were conducted in May 2021, August 2021, May 2022, and July 2022, providing four different realisations of the beach and estuary mouth morphology (Section 4.3.3). These data were complimented by open-source aerial LiDAR data, surveyed in February 2019, and single beam bathymetry data, surveyed in 2007, by the regional coastal

monitoring programme (https://coastalmonitoring.org/) to cover the estuary and supratidal areas, as well as offshore subtidal (-10 to -20 m ODN) region, respectively. Prior to developing the model domain, the various spatial data types were merged into a single 1 m x 1 m gridded spatial data set using two-dimensional linear interpolation, while ensuring that smooth elevation transitions were achieved between the different data types.

To optimise computational effort, the model grid was developed with a variable resolution. Within the embayment, the cross-

shore and alongshore resolution was fixed at 4 m, while outside the bay the cross-shore resolution was gradually decreased from 10 m depth using the Courant condition to optimise resolution based on water depth, giving a cross-shore resolution of 34 m at the offshore model boundary. Either side of the bounding headlands, the alongshore resolution was decreased from 4 m to a maximum of 20 m at the lateral boundaries. The model extends linearly 200 m to the northeast and 400 m to the southwest of the bounding headlands to ensure that any wave shadowing from the lateral boundaries does not impact conditions

within the bay, and a linear transition was implemented at the offshore boundary to remove any near-boundary gradients (Figure 5a).

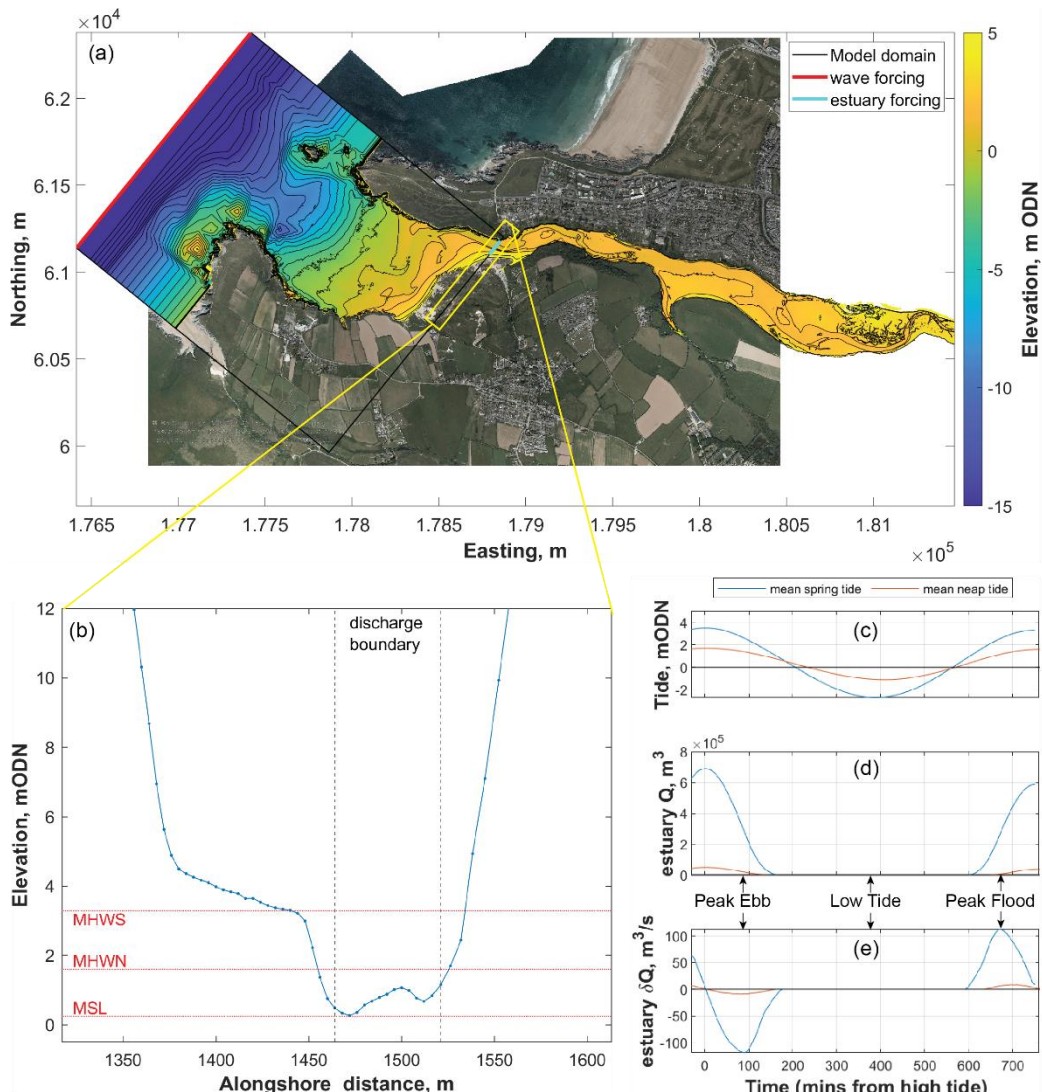

**Figure 5: (a) Model domain (black box) overlaid onto aerial imagery of Crantock Beach and surrounding area. The coloured topography shows the elevation of beach and estuary morphology. (b) Alongshore cross section of the model DEM (blue dotted line) in the area where the Gannel enters Crantock beach, showing the location (black dashed lines) and resolution (blue dots) of the discharge forcing boundary. Tidal elevations are shown for Mean High Water Spring (MHWS), Mean High Water Neap (MHWN), and Mean Sea Level (MSL). (c-e) Hypsometry of the Gannel estuary, including mean spring and neap tidal excursion in meters above Ordnance Datum Newlyn (c), submerged volume, Q (d), and change in volume $\delta Q$ (e) of the estuary. Aerial imagery courtesy of National Network of Regional Coastal Monitoring Programmes of England, © 2024 NNRCMP.**

### 3.5 Boundary forcing conditions

The XBeach model was forced with waves uniformly along the seaward boundary, while tidal variation was imposed uniformly across the modal domain. For calibration and validation, wave forcing (time-varying wave height, period, direction and

directional spread) was obtained from a nearby directional waverider buoy located 7.5 km southwest of Crantock Beach in 10 m water depth at mid-tide (https://coastalmonitoring.org/), with waves reverse shoaled to the boundary depth of 20 m using linear wave theory. Tidal variation was obtained from Admiralty tide charts. Once the model was calibrated (Section 3.7), seventy-two combinations of wave and tide conditions were selected to run in the model covering the full range of summer wave conditions (Table 1), with each set of wave conditions run over a mean neap tidal cycle and a mean spring tidal cycle

(with 30 minutes spin up time). The most energetic conditions are approximately 3.5 times higher than the summer (June, July, August) average wave power, equivalent to approximately the 1-in-1 year return period and would be conditions under which the lifeguards would close the beach to bathers. Each 12-hour simulation was then divided into 1-hour tidal segments at 30-minute increments, providing 1,728 unique combinations of wave and tide forcing from which to evaluate circulation patterns and bathing hazard from the simulated flow fields.


**Table 1. Summary of XBeach forcing variables used to populate the hazard look up table.**

| Forcing variable | Values simulated |
|---|---|
| Significant wave height, $H_{m0}$ (m) | 0.5, 1, 2, 3 |
| Peak wave period, $T_p$ (s) | 6, 9, 12 |
| Peak wave direction, $D_p$ °N (° shore normal) | 264, 279, 309 (45, 30, 0) |
| Wave directional spread, (°) | 30 |
| Tide range | mean spring tide, mean neap tide |

The XBeach model itself does not include the Gannel estuary, as, while it is possible to compute estuary flows within XBeach (Hartanto *et al.*, 2011), this would add considerable computational effort given the spatial extent of the estuary. Instead, the

ebbing and flooding flows from the Gannel were imposed using a discharge boundary. This forces a flow of water into the model domain (positive discharge) or out of the model domain (negative discharge) to describe the ebbing and flooding of the tide, respectively, through the relatively narrow estuary mouth at the north-eastern side of the beach. Discharges were applied to the model across a 60-m section of the landward model boundary, covering the deepest part of the river channel (Figure 5b). The submerged volume of the Gannel estuary ($Q;$ Figure 5d) was quantified landward of the model boundary beneath a range

of tidal elevations (Figure 5c), through interrogation of the Gannel Estuary DEM (Figure 5a). The difference in volume

between two elevation plains ($dQ$), and the timeframe over which the tide changes between those elevations ($dt$), were then used to estimate the tidal discharge into or out of the estuary at different stages of the tide ($dQ/dt$). This analysis indicates peak discharges are 8 m³/s and 110 m³/s during average neap and spring tides, respectively (Figure 5e). For the scenario simulations, the discharge applied at the boundary was computed from the estimated spring and neap tidal discharge rate at a given point

in time, plus an additional 2 m³/s to conservatively account for fluvial flow (5% exceedance river discharge) which is rare during the summer bathing season. However, initial tests with only fluvial discharge applied showed that this fluvial discharge rate has a negligible effect on surfzone flows.

## 3.6 Quantification of flow behaviour and bathing hazard

To assess surfzone circulation and quantify bathing hazard, Generalised Lagrangian Mean flow velocity (Groeneweg and

Klopman, 1998) fields from XBeach were used to advect virtual drifters within the model and the drifter tracks were then analysed to provide bathing hazard proxies. The drifters were seeded within the surfzone randomly in time and space, seeding across a depth range representing safe-depth limits for children to adults (0.7–1.2 m, respectively), informed by previous studies (McCarroll *et al.*, 2014a; McCarroll *et al.*, 2015). Depths shallower than 0.7 m are deemed 'safe' as bathers can stand up without being swept off their feet by typical surfzone currents. 1,500 drifters were seeded during each simulated period,

with 500 of these seeded along the bank of the River Gannel where the estuary enters the beach and the rest seeded along the shoreline of the beach. Each virtual drifter was advected for 20 minutes, or until they had returned to a safe water depth (< 0.7 m). The 20-minute timeframe was chosen to represent a typical timescale of a bathing incident – it is likely that a person in a strong current would either be rescued or in a critical state within 20 minutes. Furthermore, as we simulate with non-stationary tides, leaving drifters to circulate for longer blurs the effects of different tidal stages.

Seaward Lagrangian flow speed $U_{off}$ and the percentage of drifters exiting beyond the extent of the surfzone $E$ have previously been shown to provide good predictors of bathing hazard (Austin *et al.*, 2013; Scott *et al.*, 2014). We applied these parameters to the virtual drifter data from XBeach to quantify when and where peak bathing hazard occurs. Defining and determining a single value of $U_{off}$ and $E$ for each simulation is not trivial (Castelle *et al.*, 2010; Austin *et al.*, 2013), as surfzone velocities and circulation vary spatially and temporally. In rip current studies, $U_{off}$ (often termed $U_{rip}$) is usually quantified at a pre-

defined location such as the rip channel neck under study (Castelle *et al.*, 2010; Austin *et al.*, 2013), and $E$ similarly requires a specific cross-shore threshold to be crossed in order to count surfzone exits. In the present study we define $U_{off}$ following the approach of Austin *et al.* (2013) using hourly-averaged Lagrangian velocities from independent drifter passes through 10 m spatial bins:

$$U_{off} = \sqrt{\overline{u_{off}^2} + \overline{v^2}} \qquad (1)$$

where $u_{off}$ is the offshore-directed flow velocity, $v$ is the alongshore-directed velocity, and the overbar signifies time averaging. The reader is referred to Castelle *et al.* (2010) for the method of determining independent drifter passes. Only spatial bins with at least five independent drifter passes were included, and the spatial bin with the maximum $U_{off}$ value defines $U_{off}$ for the

entire beach. The $U_{off}$ values presented herein therefore represent a spatial maximum, akin to $U_{rip}$ values from previous rip current studies.

To quantify the proportion of surfzone exits $E$ occurring during each 1-hour simulated period, we determine the percentage of virtual drifters that travel seaward at least the same distance as the alongshore-averaged surfzone width. For comparability with previous studies, we define the seaward extent of the surfzone as the location where the cross-shore roller energy exceeds 10% of its cross-shore maximum (Reniers *et al.*, 2009), determining the average width of the surfzone across the length of the embayment. To forecast bathing hazard (Section 5), $U_{off}$ and $E$ were quantified at each time step across three different sections

of the beach (northern half, southern half, and estuary mouth) to acknowledge the fact that offshore flow velocity varies in different places along the shore and to differentiate the hazard a bather might experience in one part of the beach from another. However, given the large range of forcing and bathymetric combinations tested, the results presented in Section 4 summarise the variables as a single value across the entire beach.

### 3.7 Model calibration and validation

The developed XBeach model was calibrated against the first tidal cycle of Eulerian field data (Figure 6 and Figure 7), with measured and modelled wave height compared at each of the surfzone PTs, and flow velocity compared at each of the three ADV locations. Six hydrodynamic tuning parameters were adjusted in the model during calibration: the wave breaking formulation, the breaker slope coefficient, the wave dissipation coefficient, the wave breaker parameter, the bed friction formulation, and the bed friction coefficient. The wave and flow comparison was found to be insensitive to the breaker

formulation, breaker slope coefficient, and wave dissipation coefficient, so these parameters were left at their default values. The wave breaker parameter $\gamma$, which controls the break point and surfzone width, and the bed friction formulation and coefficient $C$, which influence the current velocities, had a greater effect on the hydrodynamic performance of the model. Optimal settings for these parameters were found to be a breaker coefficient of $\gamma = 0.50$ and a Chezy bed friction coefficient of $C = 45$ m$^{1/2}$/s. These same settings were previously found to provide optimal tuning for headland rip modelling by

Mouragues *et al.* (2021).

Another important calibration for the present study is the weighting of the discharge during the flooding and ebbing stages of the tide, applied at the estuarine boundary. Initially it was assumed that $dQ/dt$ could be applied directly from the hypsometric analysis (Section 3.5); while this was found to replicate flow velocities on the ebbing tide well, it overpredicted landward flow velocities on the flooding tide. Peak ebb tide velocities have been observed at other sites to occur in the deepest morphological

channels, while peak flood tide velocities are generally weaker and do not necessarily occur in the deepest channels (Allen, 1971; Lessa and Masselink, 1995). A weighting coefficient is therefore required to account for the fact that while the ebb tide exits directly through the river channel (here used as the discharge boundary), the flood tide enters the estuary through both the channel and the surrounding shoal/beach areas. Applying $dQ/dt$ across our fixed discharge boundary location (Figure 5) is therefore a reasonable representation during the ebbing phase of the tide, but not during the flood phase. To account for this,

we use separate discharge coefficients, $\alpha$, for the flood tide ($\alpha_{flood}$) and ebb tide ($\alpha_{ebb}$) periods, applied to the hypsometric discharges as:

$$(dQ/dt)\,\alpha_{flood} \qquad \text{for } \frac{dQ}{dt} \geq 0 \qquad\qquad (2)$$

$$(dQ/dt)\,\alpha_{ebb} \qquad \text{for } \frac{dQ}{dt} < 0 \qquad\qquad (3)$$

Optimal values to replicate the estuary flow velocities were found to be $\alpha_{flood} = 0.5$ and $\alpha_{ebb} = 1$. However, this may not be generalisable and would need to be calibrated for other study sites.

Using these calibration settings, the model was validated against the Eulerian data from the entire field deployment period (three tidal cycles). Model skill was quantified using the Root Mean Square Error (RMSE), defined as:

$$RMSE = \sqrt{\frac{1}{n}\sum_{i=1}^{n}\left(X_{XB(i)} - X_{M(i)}\right)^2} \qquad\qquad (4)$$

where $X$ refers to the hydrodynamic variable being assessed, and subscripts $XB$ and $M$ refer to the modelled and measured values, respectively. The RMSE values were also normalised (NRMSE) by the mean $H_s$ value or maximum $U$ value at each instrument location across the periods under comparison to assess the relative magnitude of the error (Mouragues et al., 2021).

## 4 Results

### 4.1 Eulerian and Lagrangian field measurements

The field experiment was conducted during approximately average summer wave and tide conditions ($H_s = 0.7$–1.4 m, $T_p = 7$–14 s, $D_p = 260$–290°, tide range = 5.6 m; Figure 4). Water level data from the mid-estuary PT shows that the estuary began to fill as the tide increased above ~1 mODN, with the volume peaking at approximately 500,000 m$^3$ at high tide (Figure 6). Estuary flow, however, peaked just over an hour before high tide (landward discharge) and after high tide (seaward discharge), with fluxes of up to 100 m$^3$/s estimated from the water levels measured during the experiment (Section 3.2).

The estuary discharge generated strong surfzone flows, which can be seen in the observed field data. Figure 6 demonstrates Eulerian velocities from the ADVs located on the intertidal ebb shoal bank adjacent to the estuary mouth, and within the estuary mouth itself. Each of the measured tidal cycles showed a similar hydrodynamic signature: Peak velocities in the estuary mouth occurred 1–2 hours before and after high tide and diminished to zero at high tide. Comparable velocities were measured on the ebb shoal bank, 30 minutes after (before) the estuary mouth on the rising (falling) tide. The seaward flow out of the estuary measured on the falling tide was 1.5–2 times faster than the landward flow on the rising stage of the tide, with seaward velocities at the ADV peaking at 0.75–1 m/s and landward velocities at 0.5 m/s for these two example tides. However, as the ADV in the estuary mouth was not in the deepest part of the channel, it is expected that peak flows exceeded 1 m/s elsewhere. The intertidal beach face ADV shows velocities remained notably low on the rising and high stages of the tide but increased rapidly up to 1.5 m/s on the falling tide especially once the water level had dropped below 1.5 mODN, concentrating the flow within the submerged beach face river channels. This ADV was located in the centre of the channel, so this represents a realistic

estimate of the peak ebb tide velocity. The *u* and *v* velocity components (Figure 7) indicate that cross-shore flows dominated within the seaward-facing estuary mouth, while alongshore flows dominated on the ebb-shoal bank and within the alongshore-oriented beach face river channel.

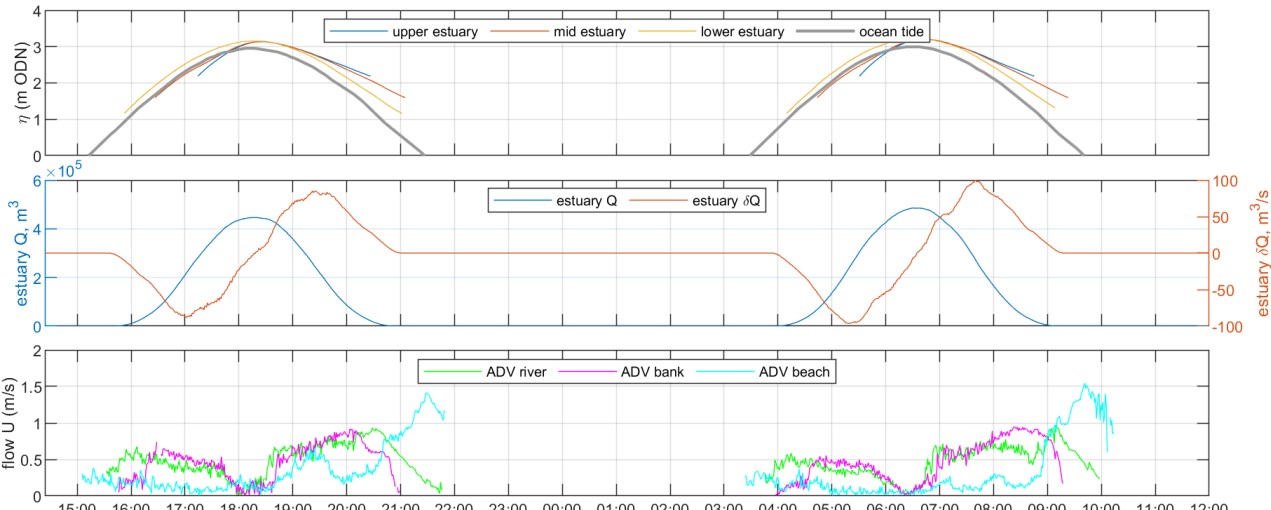

**Figure 6: Eulerian field data collected in the vicinity of the Gannel estuary over the first two high tides of the field experiment. Upper panel: water level signal measured by PTs at different locations in the Gannel estuary and ocean tide level from Admiralty Tide Charts. Middle panel: Gannel estuary volume, *Q*, and change in volume, *dQ*, estimated from the mid estuary PT data, as per Section 3.5. Lower panel: velocity magnitude measured by ADVs in the estuary mouth ('ADV river'), ebb shoal river bank ('ADV bank'), and within the beach face channel (ADV beach). See for instrument locations. Note that the estuary is dry 3 hours either side of low tide.**

The Lagrangian GNSS drifter tracks reveal that various circulation behaviours occurred simultaneously across the beach, including alongshore, rotational, and exiting flows, creating a complex flow field (Figure 8). The observed circulation patterns evolved from low to high tide due to differences in the underlying beach morphology, as well as the activation and de-activation of the estuary at mid-tide. The GNSS drifters show a strong estuarine current flowing seaward during the ebbing high tide (Figure 8, lower panel), which diverts laterally across the beach through various submerged river channels. Median Lagrangian velocity observed during this phase of the tide was 0.5 m/s, but drifter velocities exceeding 1 m/s occurred where the river channel bends sharply away from the northern headland. Although many of the GNSS drifters exited the surfzone during the ebbing phase of the high tide, rotational flow also returned some drifters shoreward and back along the shore towards the main river channel (Figure 8). Conversely, at low tide the GNSS drifters show weaker onshore and alongshore flows leading to boundary rips at either headland, which exit seaward or circulate back towards shore (Figure 8, upper panel). The spatially-averaged lagrangian velocity during this phase of the tide was 0.3 m/s, with peak velocities exceeding 0.6 m/s within the surfzone and in the neck of the boundary rip channels. Apart from a shallow (< 1 m), fast flowing fluvial component, the estuary was inactive at tide levels below ~0 mODN.

## 4.2 Comparison of modelled and measured flows

Comparing the measured and modelled Eulerian flows (Figure 7), the XBeach model reproduces the flow magnitude and phase from the ADVs well, albeit with some underprediction of flow during the ebbing tide on the ebb-shoal bank and within the alongshore-oriented beach face river channel. The model reproduced the direction of the Eulerian flows well, with cross-shore velocities dominating in the river channel, and alongshore velocities dominating on the ebb-shoal riverbank and in the alongshore-oriented beach-face river channel. Overall, the calibrated model achieved RMSE (NRMSE) of 0.2 m (20%) for $H_s$, 0.16 m/s (20%) for $u$ velocity, and 0.22 m/s (21%) for $v$ velocity. Bias (Mean Absolute Error) in the flow magnitude was -0.07 m/s, indicating the model tends to slightly underpredict the flow velocities.

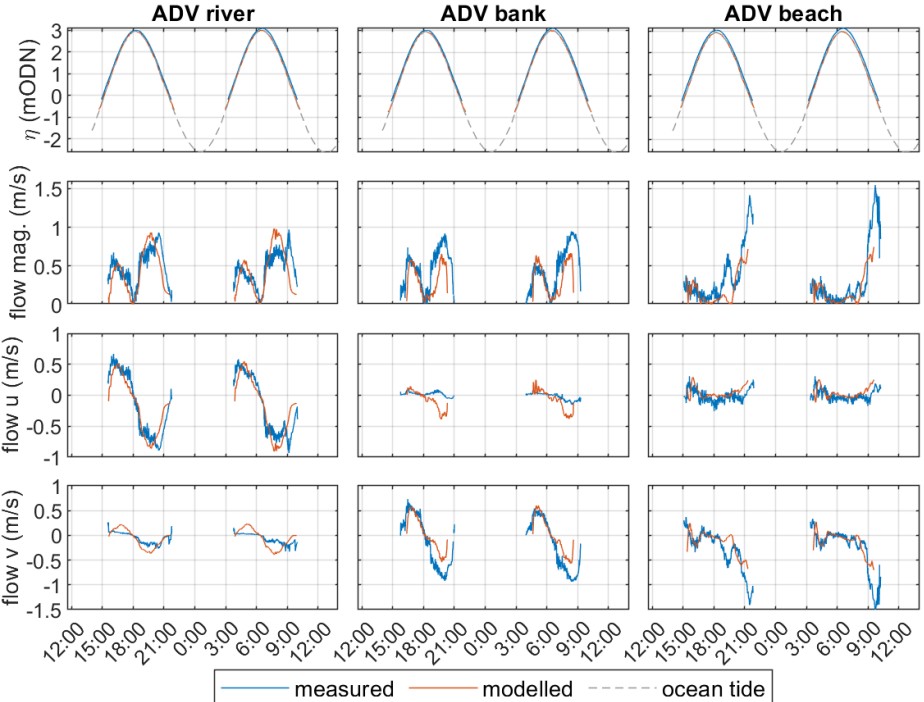

**Figure 7: Comparison between measured and modelled flow velocity at the three Acoustic Doppler Velocimeters (ADV) during the first two tidal cycles of the field experiment used for model calibration. The location of each ADV is shown in Figure 2. Positive (negative) $u$ velocities represent landward (seaward) flows, while positive (negative) $v$ velocities represent northward (southward) flows.**

Compared to the Lagrangian circulation patterns measured in the field during the low tide period (Figure 8, upper panels), the bin-averaged velocities from the virtual drifters (Section 3.6) reproduce the measured boundary rip velocities at the north and south headlands (~0.5 and ~0.7 m/s at the southern and northern headlands, respectively). The virtual drifters also capture the net onshore flow in the middle of the bay and net seaward flows in the southern half of the bay and adjacent to the north headland. During the high ebb-tide period (Figure 8, lower panels), both the measured and modelled drifters coherently follow the river channels in the beach face, with seaward flows diverting away from the north headland across the beach before either

exiting seaward through the beach face river channel, or circulating back towards the estuary along the shore. Interestingly, large numbers of virtual drifters are predicted to pass through certain points in the inner surfzone (Figure 8, panel e), which are interpreted as stagnation points where quiescent circulation between onshore and offshore flow is occurring. These features are not present in the real GNSS drifter tracks because of the statistical limitations of deploying a small fleet of drifters, but the velocity patterns in these areas are reproduced. The model reproduces the strong seaward ebb-tide flow entering the beach

from the estuary, where median Lagrangian velocities exceeding 1 m/s were both modelled and observed. The circulation patterns show that the ebb-tide flows convey water rapidly through the main river channel before they connect to channel rips and boundary rips 100's of meters away from the estuary mouth.

Overall, the calibrated model reproduces the measured Eulerian and Lagrangian flows from the three-day field deployment well, especially considering the complexity of the flow field, and is therefore deemed suitable to assess surfzone circulation

under a wider range of conditions.

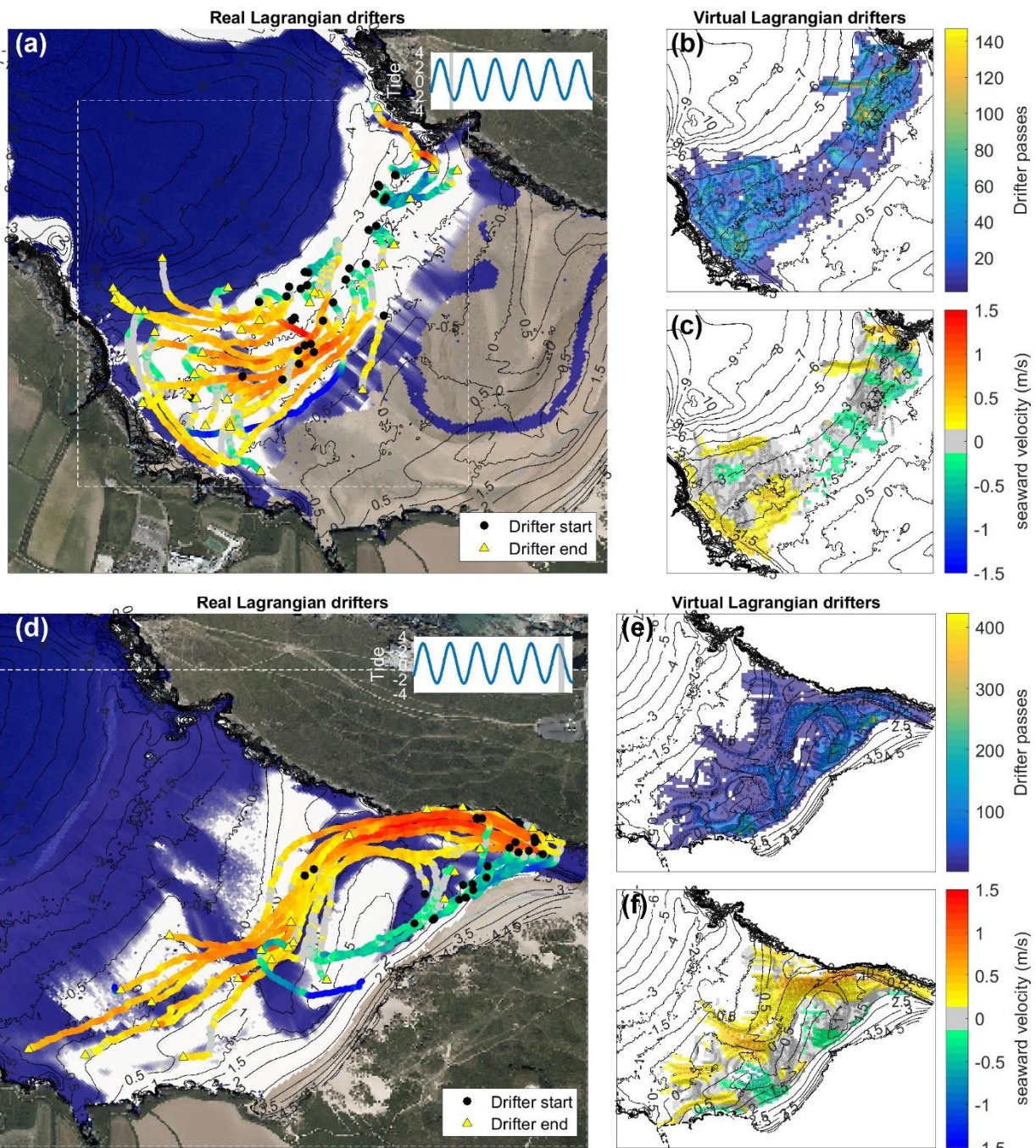

**Figure 8: Comparison of measured and modelled circulation patterns using Lagrangian velocities from real GNSS drifters (panels a and d), and virtual drifters seeded in the XBeach model (panels b, c, e, f). Panels a–c show low tide drifter data from the 12th of May 2021 and panels d–f show high ebb-tide drifter data from the 14th of May 2021. White dashed boxes in the left panels indicate the zoom extents of the right panels. Circles in the right panels show example virtual drifter tracks while colours represent drifter**

velocities averaged onto 10 m spatial bins. Blue and white areas in the left panels show average water level and predicted wave dissipation > 1 W/m² during the field experiment, for reference. Aerial imagery courtesy of National Network of Regional Coastal Monitoring Programmes of England, © 2024 NNRCMP.

### 4.3 Simulated Lagrangian circulation

In the following sections, the calibrated XBeach model is used to explore circulation patterns under a wider range of wave and tide conditions than was achieved in the field, as well as quantifying the exit potential and flow velocities under different forcing scenarios.

#### 4.3.1 Influence of tidal stage and wave conditions

The XBeach simulations demonstrate that tidal translation across the beach significantly alters the surfzone circulation. Different circulation patterns and associated bathing hazard are seen at high, mid, and low tide, and, furthermore, circulation is predicted to be distinctly different under a rising tide or a falling tide. For example, during a high ebbing spring tide with average waves (Figure 9a), the virtual Lagrangian drifters are carried alongshore by littoral currents before exiting seaward in a narrow and fast flowing estuarine current ($U_{off}$ = 1.1 m/s; $E$ = 56%). At mid-high tide stages (Figure 9b, c, f, g), the surfzone is wider and the flows become channel-constrained under average waves. The flows follow the path of the submerged river channels alongshore and seaward towards channel rips and headland boundary rips, which are enhanced when the tide is ebbing ($U_{off}$ = 1.2 m/s; $E$ = 25%) and diminished, but still present, when the estuary is flooding ($U_{off}$ = 0.52 m/s; $E$ = 0%). Below mid-tide (Figure 9d, e), the river channels and estuary discharge no longer influence surfzone circulation, and seaward flows predominantly occur within headland boundary rips at either side of the beach under average waves ($U_{off}$ = 0.59 m/s; $E$ = 13%). Tide range influences the hazard signature in two distinct ways. Firstly, it determines how much water is flushed from the estuary during the ebbing high tide phase, with an order of magnitude more estuary discharge (Figure 5) during a spring tide ($Q \approx 700$ m³) than during a neap tide ($Q \approx 50$ m³), which substantially increases the estuary-driven flows in the hour following high spring tide ($U_{off}$ = 1.1 m/s; $E$ = 56%) compared to those following high neap tide ($U_{off}$ = 0.63 m/s; $E$ = 15%). However, at equivalent tide elevations, the circulation patterns are almost identical during a neap and spring tide (not shown here) because the ebb shoal channels dictate the circulation. Secondly, during high (Figure 9a) and low (Figure 9e) spring tides, the beach gradient is steeper and the surfzone is narrower than at any stage of a neap tide. This controls the ability of estuary and rip current flows to exit the surfzone, with a far wider, more saturated surfzone, and subsequently fewer exits, occurring during a neap tide than under high or low spring tides.

Figure 10, panels a-b, summarise the predicted flow velocity and exit potential under various combinations of wave and tide conditions. Wave power is parameterised relative to the timeseries mean using a 'wave factor' parameter (Scott *et al.*, 2014) $W_f = H_s T_P / (\overline{H_s T_P})$, which describes the ratio between the associated $H_s T_P$ (proportionally representing wave power) and the 16-year summer (June, July, August) mean ($\overline{H_s T_P}$). This analysis shows that below mid-tide, tidal direction (falling or rising)

becomes relatively unimportant, but seaward flow and exit potential are predicted to increase with increasing wave power, from $U_{off}$ = 0.3–0.5 m/s and $E$ = 5–15% when $W_f < 0.5$, to $U_{off}$ = 0.7–1.0 m/s and $E$ = 20–40% when $W_f$ = 3–4. However, above mid-tide there are significant differences predicted in flow velocity and exit potential under rising or falling stages of the tide. The least hazardous conditions are predicted to occur during a rising mid to high tide (Figure 10, panel a) with wave power below average, when seaward flows are almost entirely absent ($U_{off} < 0.5$ m/s; $E$ = 0%). Conversely, the most hazardous

flows are predicted to occur during ebbing high spring tides (Figure 10, panel b) with wave power below average ($U_{off}$ = 1.5 m/s; $E$ = 58%), when the estuary discharge can flow seaward completely unhindered by the surfzone. As wave power increases above average ($W_f > 1$), these ebbing flow velocities are predicted to remain high, but exit potential decreases significantly, as estuary flow is hindered by a wider, more saturated surfzone. This is further summarised in Figure 10, panels c-d, which compare variation in $U_{off}$ and $E$ over a mean spring tide for four different $W_f$ values, with a fixed wave period and direction of

$T_p$ = 12 s and $D_p$ = 279°. This demonstrates that during high ebbing tides, the highest exit potential and offshore flow speeds occur when wave power is low, while during all other stages of the tide, high wave power leads to increased exit potential and flow speeds.

Wave direction also appears to play a role in controlling the hazard signature at Crantock. Considerably more surfzone exits were predicted at the southern 'downstream' headland of Crantock (max $E$ = 90%) or at the northern 'upstream' headland

(max $E$ = 72%) depending on the angle of wave approach. Below mid tide, wave direction varied $U_{off}$ by only 0.008 m/s on average, but $E$ increased by 12% when wave direction was varied from the most oblique wave approach simulated (45°) to a shore-normal wave approach (0°). Figure 10, panels e-f, compare variation in $U_{off}$ and $E$ over a mean spring tide for three different wave approaches, with a fixed wave height and period of $H_s$ = 1 m and $T_p$ = 12 s ($W_F$ = 1). This shows that drifters released in the north of bay had a much lower exit potential than drifters released in the south of the bay during oblique wave

approaches, due to the southern drifters being released near the shadow boundary rip that occurs under oblique waves. Despite starting in the northern half of the beach, drifters released in the Gannel estuary mouth had a high exit potential even during oblique waves, as the estuary flows transported drifters rapidly south towards the southern headland boundary rip. The ebbing estuary flows are predicted to be hindered during shore-normal waves with average wave power, with exit potential and seaward flow speed predicted to be reduced by approximately half compared to during oblique wave approaches.

Overall, the simulations suggest that while wave power and direction strongly influence the hazard signature, tidal stage plays the most important role in controlling both surfzone exits and the velocity of surfzone currents at an embayed, estuary mouth beach like Crantock, with $U_{off}$ and $E$ varying by up to 0.44 m/s and 70%, respectively, when averaged at each tidal level.

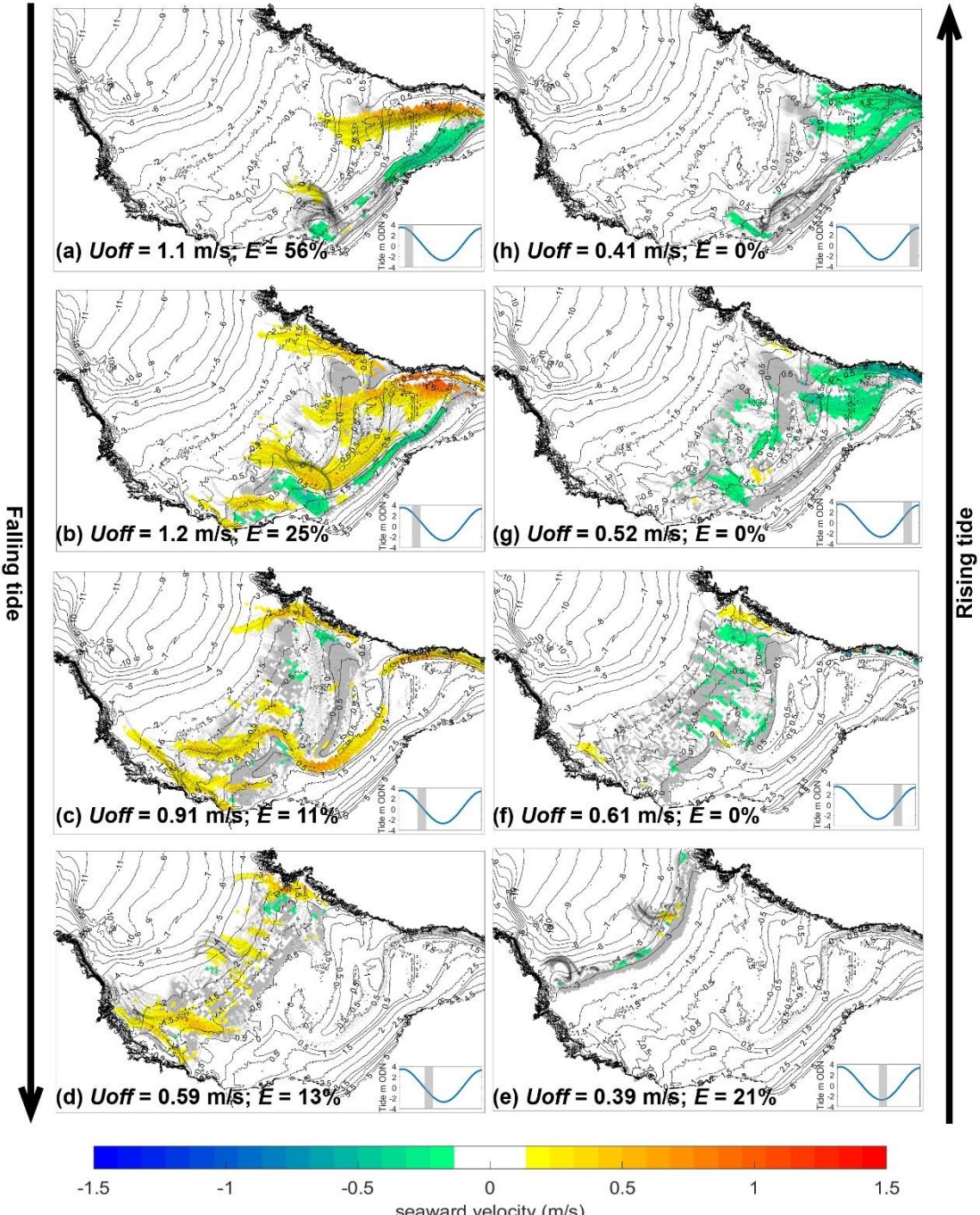

**Figure 9: Simulated Lagrangian surfzone circulation from XBeach during a spring tidal cycle with average wave conditions ($H_s = 1$ m, $T_p = 12$ s, $D_p = 279°$, $W_F = 1$). Circles show example virtual drifter tracks while colours represent drifter velocities averaged over 1 hour onto 10 m spatial bins. The bin with the maximum seaward directed flow velocity defines $U_{off}$ for the entire beach, while $E$ is computed from the proportion of virtual drifters that exit the surfzone (Section 3.6). Contours indicate beach morphology (m ODN) while grey areas show where breaker dissipation exceeds 10% of the surfzone maxima. Tidal stage is shown in the inset panels.**

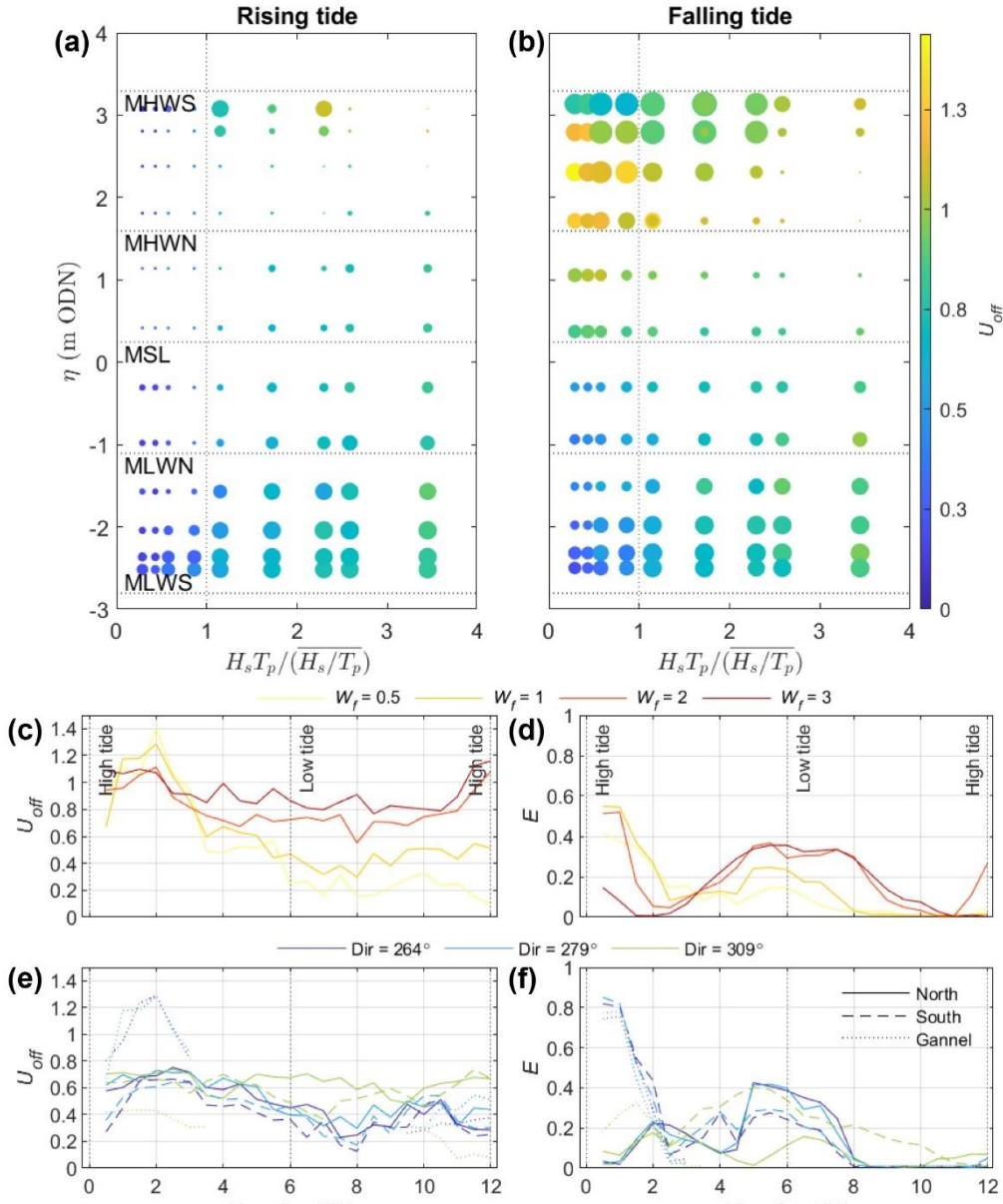

**Figure 10: (Panels a-b)** Proportion of surfzone exits ($E$, bubble size) and offshore-directed drifter velocity ($U_{off}$ m/s, bubble colour) from XBeach simulations with a mean spring tide and mean angle of wave approach ($D_p = 279°$). Bubble size range represents $0 \le E \le 58\%$. Results are plotted as a function of water level $\eta$ and wave factor $W_f = H_s T_p / (\overline{H_s T_p})$. The vertical dashed line indicates average wave power $(\overline{H_s T_p})$, while the horizontal lines represent Mean High Water Spring (MHWS), Mean High Water Neap (MHWN), Mean Sea Level (MSL), Mean Low Water Neap (MLWN), and Mean Low Water Spring (MLWS) water levels. **(Panels c-d)** variation in $U_{off}$ and $E$ over a tidal cycle for four different $W_f$ values, with a fixed wave period and direction of $T_p = 12$ s and $D_p = 279°$. **(Panels e-f)** variation in $U_{off}$ and $E$ over a tidal cycle for three different wave directions, with a fixed wave height and period of $H_s = 1$ m and $T_p = 12$ s ($W_f = 1$). Solid and dashed lines compare results for drifters released from the northern and southern half of Crantock beach, while dotted lines show drifters released along the Gannel estuary mouth (only for tide > 1 mODN).

### 4.3.2 Influence of estuary discharge

The estuary has been shown to drive extremely strong seaward flows through the surfzone in the hours following a high spring tide, with flow velocities up to 1.5 m/s. However, the estuary influences the flow dynamics in two distinct ways: (1) water flushing into and out of the estuary drives increasingly strong surfzone currents as tide range increases; and (2) the river channel morphology acts to constrain surfzone flows in the same way that rip channels funnel wave-driven flows. In fact, ignoring estuary flows, the ebb shoal delta acts very much like a bar-rip system found on an intermediate morphology beach, providing shallow areas that induce wave breaking and deeper areas where wave-driven flows can return seaward. To demonstrate this, XBeach simulations were compared with and without estuary discharge activated in the model, to disentangle the effect of the river channel morphology from the strong estuary flows (Figure 11).

In the hours immediately prior to a high spring tide a significant flow of water floods the estuary (Figure 11a), with strong landward flows occurring near the shore adjacent to the estuary. With estuary discharge switched off in the model (Figure 11c), onshore flows are still predicted to occur near the shore due to wave breaking over the shallow ebb-shoal bathymetry, but the strong landward flows in the estuary mouth no longer occur. However, wave breaking on the ebb shoal delta is predicted to drive weak seaward flowing rips in the river channel away from the estuary, especially when estuary discharge is switched off, indicating that the river channel morphology induces channel rip behaviour independently of the estuary flows.

Immediately following a high spring tide there is a strong seaward flow driven by the estuary (Figure 11b). Switching the estuary discharge off in the model (Figure 11d) unsurprisingly has a strong effect on the flows immediately adjacent to the estuary mouth, but in the river channels more than 300 m from the estuary mouth the predicted circulation patterns are very similar whether estuary discharge is applied in the model or not. The river morphology is predicted to induce channel rip behaviour and contributes to bathing hazard independently from the estuary flows. Seaward flows in the main river channel in the middle of the beach are predicted to be similar in magnitude (0.6 m/s) in the absence of estuary flows, to those with estuary flows activated (0.8 m/s), as are the boundary rip current velocities (0.8 m/s).

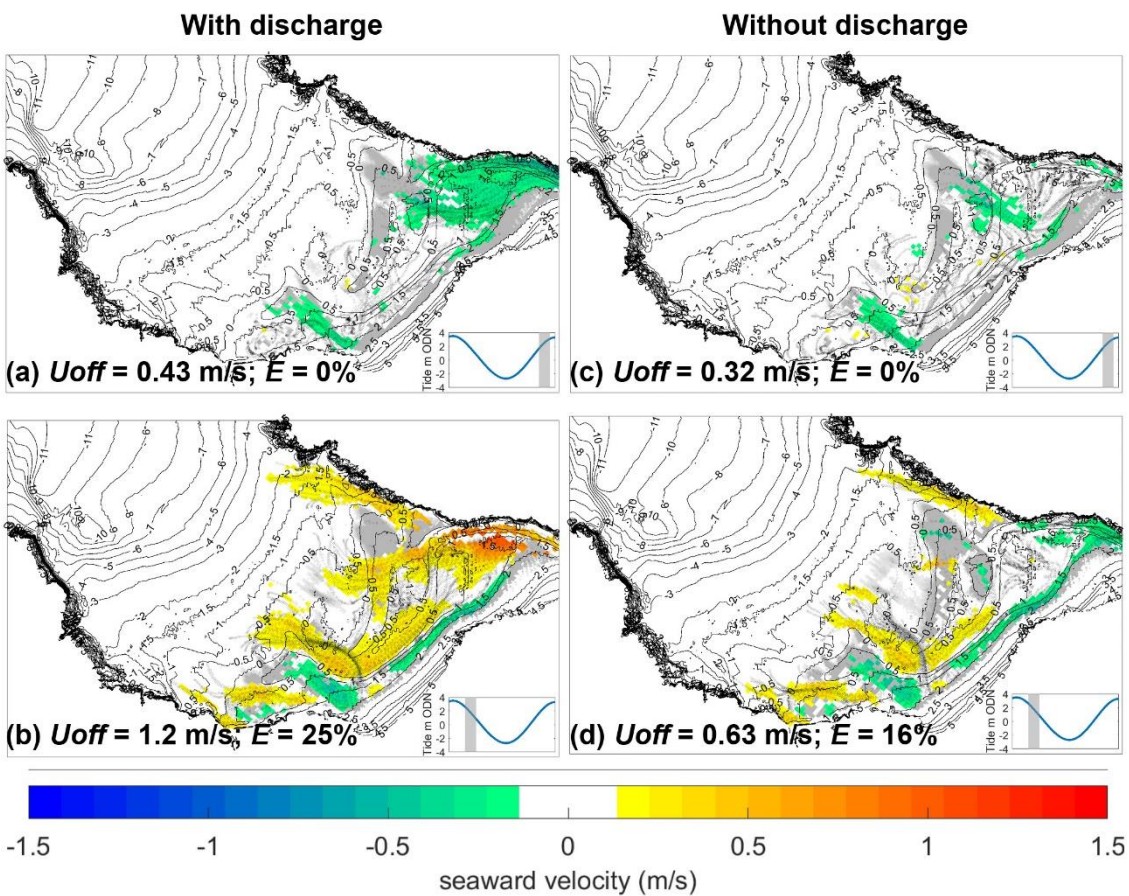


**Figure 11: Simulated Lagrangian surfzone circulation from XBeach with (left panels) and without (right panels) estuary discharge activated in the model. The simulated conditions are 1.5–2.5 hours before (upper panels) and after (lower panels) a high spring tide with average wave conditions ($H_s$ = 1 m, $T_p$ = 12 s, $D_p$ = 279°). Circles show example virtual drifter tracks while colours represent drifter velocities averaged over 1 hour onto 10 m spatial bins. The bin with the maximum seaward directed flow velocity defines $U_{off}$**
**for the entire beach, while $E$ is computed from the proportion of virtual drifters that exit the surfzone (Section 3.6). Contours indicate beach morphology (m ODN) while grey areas show where breaker dissipation exceeds 10% of the surfzone maxima. Tidal stage is shown in the inset panels.**

### 4.3.3 Influence of beach morphology

Four different surveys of the sub- and inter-tidal bathymetry of the beach were conducted during May 2021, August 2021, May 2022, and July 2022 (Section 3.4). As Figure 12 shows, the morphology of the upper intertidal beach is dominated by the main river channel, which carves troughs >1 m deep in the beach face and evolves noticeably over the four surveys. In contrast, the lower beach contours remain relatively stable over this period, aside from slight variations in the ~1 m deep headland boundary rip channels at either side of the beach, and the appearance and disappearance of smaller rip channels away from the
headlands. The main river channel enters the beach with a constant position along the northern headland before deflecting

away from the headland towards the middle of the beach. Initially, this channel exits approximately through the middle of the beach face (May 2021; Figure 12a), but in subsequent surveys the river channel is seen to shift periodically a few hundred meters southward (Aug 2021; Figure 12b) and northward (May 2022; Figure 12c). Less defined channels are also exhibited at various points in time, revealing relic positions of the main river channel.

XBeach simulations performed on the four different realisations of the morphology show that the variation in the spatial flow patterns over the four surveys is quite substantial (Figure 12, a-d), with the direction of the flows through the main river channel varying by more than 45 degrees across the middle of the beach. Furthermore, the precise position of those strong channelised flows shifts over a distance of a few hundred meters. However, the hazard signature, here represented by $U_{off}$ and $E$, is altered surprisingly little by the variations in the morphology (Figure 12, e-f). Over an average spring tide with average wave forcing

($H_s = 1$ m, $T_p = 12$ s, $D_p = 279°$), analysis of the virtual drifters suggests that the hazard signature is consistent across the four bathymetries, with seaward flows and surfzone exits maximised within 2 hours of high tide ($1 < U_{off} < 1.4$ m/s; $30\% < E < 60\%$) and reduced seaward flows ($0.3 < U_{off} < 0.6$ m/s) predicted for the remainder of the tidal cycle, albeit with a second peak in exits occurring at low tide ($E \approx 20\%$). For a given tide level, the standard deviation in $E$ and $U_{off}$ across the four bathymetries is $\leq 8\%$ and $\leq 0.22$ m/s, respectively, with maximum differences of up to 18% and 0.53 m/s, respectively. In comparison, the

variation in $E$ and $U_{off}$ due to the tide moving over a single bathymetry is 60% and 1.2 m/s.

This suggests that temporal variation in the position and flow direction of the river channels due to the shifting morphology does not significantly impact the overall hazard characteristics, and that for a given set of forcing conditions the variation in morphology plays a secondary role in the hazard level compared to the effect of varying the forcing conditions themselves. This relative insensitivity of the hazard level on beach morphology facilitates the development of a beach hazard prediction

tool (Section 5), as frequent morphological updating is not required.

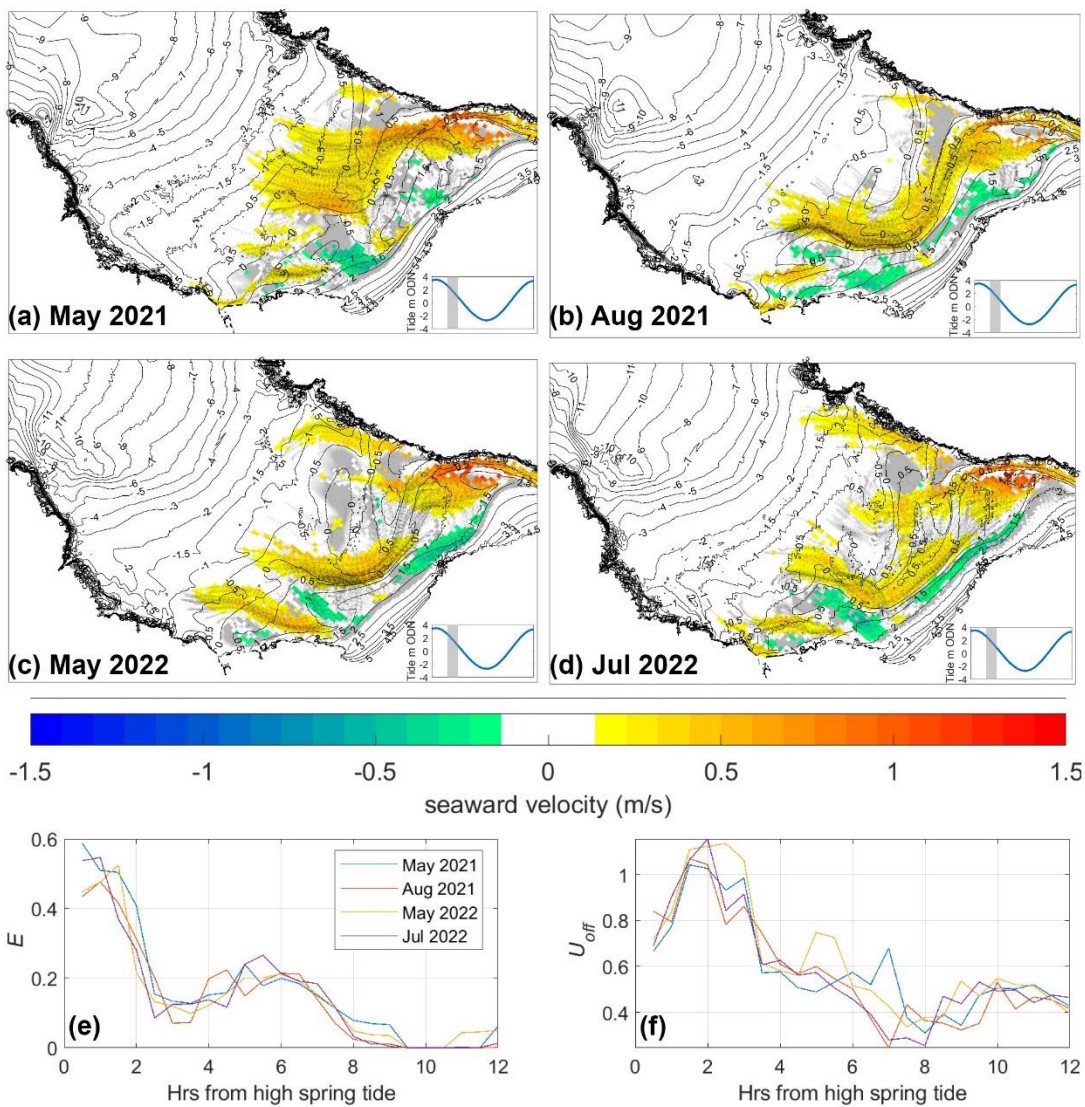

**Figure 12**: (Panels a-d) Lagrangian surfzone circulation from XBeach run with four different measured bathymetries. The simulated conditions are 1.5-2.5 hours after a high spring tide with average wave conditions ($H_s = 1$ m, $T_p = 12$ s, $D_p = 279°$). Circles show example virtual drifter tracks while colours represent drifter velocities averaged over 1 hour onto 10 m spatial bins. Contours indicate beach morphology (m ODN) while grey areas show where breaker dissipation exceeds 10% of the surfzone maxima. Tidal stage is shown in the inset panels. (Panel e) proportion of surfzone exits $E$ and (Panel f) seaward flow velocity $U_{off}$ predicted by XBeach (Section 3.6) over a spring tidal cycle with average wave conditions over the four different bathymetries.

## 5 Forecasting bathing hazard

A site-specific bathing hazard forecast was developed for Crantock Beach in collaboration with the local landowners and the RNLI, with the intention of informing beach users of hazardous currents prior to accessing the beach. The calibrated XBeach model was used in the forecast system by running a suite of offline simulations (described in Section 3.5 and Table 1) and storing $U_{off}$ and $E$ values alongside their associated forcing conditions from each hourly segment of model output in a look-up table. The simulations used to populate the database were carefully designed to cover the full range of summer wave and tide

conditions, while also optimising the computational effort required to initially populate the database (Section 3.5). Forecasted ocean wave and tide conditions at the boundary of the XBeach model are gathered each day from the UK Met Office AMM15 wave and tide models. These are compared using a nearest-neighbour search to the forcing conditions in the look-up table, to find the nearest simulated conditions to those forecasted in the coming days, resulting in a timeseries of predicted $U_{off}$ and $E$ values. A 'Hazard Score' (HS1, HS2, or HS3) is then applied to each forecast timestep by comparing the predicted $U_{off}$ and $E$

values for that timestep to pre-determined thresholds calibrated below (Table 2). Low values of both $U_{off}$ and $E$ represent the lowest bathing hazard (HS1), as bathers would be advected slowly and retained in shallow water, while high values of $U_{off}$ and $E$ represent the highest hazard (HS3) as bathers would be transported quickly towards deep water. Other combinations represent a medium hazard level (HS2).

To calibrate the hazard thresholds, hazard must first be quantified in some way using records of past bathing incidents.

Applying the approach endorsed by the World Health Organisation (WHO) for emergency and disaster risk management (Saulnier *et al.*, 2020), it can be said that the total number of bathing *Incidents* (or in WHO terms, the 'Life Risk') that occur over a given period is the product of three key factors: '*Exposure*', '*Hazard*' and '*Vulnerability*' (Kennedy *et al.*, 2013). For example, a high number of bathing incidents can occur at a beach even when modest hazards are present, if the number of water users is high, or if those water users are particularly vulnerable to the hazards due to low water competency or low hazard

awareness. Conversely, if only a few people of average vulnerability enter the water on a given day but each one of those people gets caught in a current and needs to be rescued, the number of incidents would be relatively low but the hazard can be considered high as the probability of each water-user being in an incident approaches unity. Assuming it is informed by a suitably large number of observations, this relationship between past incidents and hazard level can be simplified to:

*Hazard = Incidents/Exposure* (1)

This parameterisation of *Hazard* represents the probability of an individual water-user of 'average' vulnerability (i.e. average swimming ability and surf-zone competency) being involved in a flow-related incident over a given time frame, and has been applied in previous studies to define hazard at the coast (Scott *et al.*, 2014; Stokes *et al.*, 2017; Castelle *et al.*, 2019).

The thresholds in Table 2 were optimised by analysing past bathing incidents at Crantock Beach over the years 2016–2021. Only flow-related incidents (n = 648) were considered where a lifeguard was required to rescue or assist a water-user back to

shore (Figure 13). The lifeguard data were discretised into 2-hour time bins and the number of *Incidents* were divided by the bather head count made by the lifeguards during each 2-hour period (representing an estimate of the average *Exposure* over

that period), resulting in an 'observed' *Hazard* level from Eq. 1 for each timestep. The *Hazard* timeseries was then used to compute bin-averaged *Hazard* values across a number of discrete $U_{off}$ and *E* bins. The distribution of *Hazard* over these bins suggests that sharp increases in *Hazard* occur when $U_{off}$ reaches 0.2 m/s and 0.4 m/s. The lower threshold is corroborated by

Moulton *et al.* (2017a), who identified that rip current speeds greater than 0.2 m/s may be hazardous to swimmers. For *E* we find a single threshold of 0.2 (20% likelihood of a drifter exiting the surfzone), which distinguishes between lower and higher levels of *Hazard*. An obvious second increase in *Hazard* with E was not visible from the distribution. Using these thresholds, two scores are obtained from Table 2 which are added together and rounded to achieve a final Hazard Score, following the approach of Austin *et al.* (2013).

To assess the skill of the developed forecast system we consider how often the upper Hazard Scores (HS2 and HS3) were hindcasted when an incident was recorded by the lifeguards. This is termed the Probability of Detection (Panofsky and Brier, 1965), also known as the Recall or Sensitivity, of the predictive system and represents the rate of true positives achieved. We also examine how often the hindcast missed an incident (i.e. HS1 was predicted when an incident occurred) which represents the rate of false negatives. It is not possible to examine the rate of false alarms ('false positives') because bathing hazard can

be high without an incident occurring, for example if no one enters the water or due to lifeguard preventative actions.

The highest hazard scores (HS2 and HS3) were forecasted most often (65% and 26% of the time, respectively) but also captured most of the observed *Incidents* and *Hazard*. True positives (false negatives) were achieved 98% (2%) of the time. HS1 was forecasted least often (8%) but satisfactorily captured only 2% of the observed *Incidents*, with an average *Hazard* probability of 1 incident per 3,303 bathers (Figure 13) and therefore represents a low *Incident*, low *Hazard* scenario. HS2 represents a

high *Incident*, medium *Hazard* scenario, capturing 57% of past incidents with an average *Hazard* probability of 1-in-1,227. HS3 represents a high *Incident*, high *Hazard* scenario, capturing 39% of past incidents with an average *Hazard* probability of 1-in-770. The likelihood of an individual bather being involved in an incident therefore increases by 2.7 times between HS1 and HS2 and by 5.7 times between HS1 and HS3. Interestingly, *Exposure* is almost equal at each hazard level, indicating that water-users are either knowingly entering the water during hazardous conditions, or more likely, are unaware of the higher

hazard occurring at certain times. The developed predictive system appears to be able to differentiate periods of low and high bathing hazard with a high Probability of Detection of recorded incidents and is therefore deemed suitable for use in the predictive forecast system at Crantock Beach to forewarn bathers of hazardous conditions (Figure 14).

**Table 2. Hazard thresholds applied to seaward flow velocity $U_{off}$ and proportion of exits $E$ from each model simulation to calculate bathing Hazard Score.**

| $U_{off}$ (m/s) | | $E(\%)$ | |
|---|---|---|---|
| Threshold | Score | Threshold | Score |
| <0.2 | 0.5 | <0.2 | 0.5 |
| 0.2-0.4 | 1 | ≥0.2 | 1.5 |
| >0.4 | 1.5 | | |

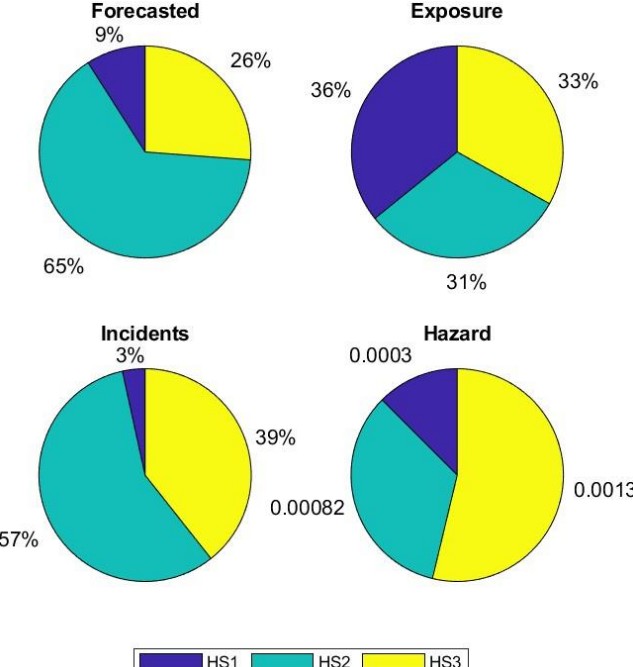

**Figure 13: Performance summary of the developed bathing hazard forecast over the hindcast period (2016–2022). Proportion of forecasted Hazard Scores (HS1, HS2, HS3; upper left), relative average water-user exposure (upper right), proportion of total incidents (lower left), and probability of an individual water user being in a flow-related incident (*Hazard*, lower right).**

## 6 Discussion

6.1 Comparison to rip current hazards

It is well known that strong flows can occur in estuaries, and they have previously been identified as hazardous locations for bathing, for example in Goa, India, by Chandramohan *et al.* (1997). In this study it has been shown that the presence of a large ebb delta on a macrotidal beach can lead to powerful and spatially complex surfzone flows. At Crantock the estuary is inactive below mid-tide, at which point surfzone currents consist of onshore and alongshore wave driven flows and seaward flowing

boundary rip currents (for example, Figure 8, upper panel), which exhibit velocities that are typical of rips observed globally

in other studies. For example, under a range of different wave conditions we predict wave-driven rip velocities of $U_{off}$ = 0.5–1 m/s (Figure 10) during the lower half of the tide, which is in line with Lagrangian velocities of 0.4–1 m/s measured at microtidal, mesotidal, and macrotidal bar-rip beaches around the world (Austin *et al.*, 2010; MacMahan *et al.*, 2010; Scott *et al.*, 2014; McCarroll *et al.*, 2017; Moulton *et al.*, 2017a; McCarroll *et al.*, 2018). Similarly, the proportion of surfzone exits per hour that we predict when the estuary is inactive is on average $E$ = 15% (range $E$ = 0–42%), in line with global field and modelling

studies of morphologically-controlled rip currents that find average exits of $E$ = 14–19% (range $E$ = 0–34%) (MacMahan *et al.*, 2010). Therefore, when the estuary is inactive, the bulk characteristics of the rip currents at this beach are no different from those observed at other beaches globally.

When the estuary is ebbing, however, the time-averaged surfzone velocities we measured and modelled reached $U_{off}$ = 1.5 m/s, which is ≥ 50% faster than velocities typically measured in channel rips (Austin *et al.*, 2010; MacMahan *et al.*, 2010;

Scott *et al.*, 2014; McCarroll *et al.*, 2017; McCarroll *et al.*, 2018). Even boundary rip currents typically exhibit Lagrangian velocities below 1 m/s (Castelle and Coco, 2013; McCarroll *et al.*, 2014b), even under high-energy waves (Mouragues *et al.*, 2020; Mouragues *et al.*, 2021).

## 6.2 Embayment, estuary, and wave controls on surfzone exits

The exit potential during the ebbing phase of the tide ($E$ ≤ 62%) is high, but is in line with observed and modelled wave-driven

boundary rips from embayed beaches of a similar size (Castelle and Coco, 2013; McCarroll *et al.*, 2014b). The predominant oblique wave approach results in alongshore varying wave exposure, driving an alongshore current towards the south headland where it deflects offshore. We find that considerably more surfzone exits occur either at the southern 'downstream' headland of Crantock (max $E$ = 90%) or at the northern 'upstream' headland (max $E$ = 72%) depending on the angle of wave approach, in line with previous findings for embayed beaches (Castelle and Coco, 2013; McCarroll *et al.*, 2014b).

During the upper half of the tide, surfzone exits are maximised when wave power is below average, allowing the estuary to ebb unhindered by waves, and exits decrease when wave conditions are more energetic than average ($W_f$ > 1). When the estuary is flooding, surfzone exits cease completely, regardless of the level of wave energy. Surfzone exits at an estuary mouth beach are therefore strongly controlled by both wave and estuary processes. While landward flows on the flood tide intuitively reduce the likelihood of material exiting the surfzone, waves breaking seaward of the ebb shoal delta under neap high tides

and/or high wave energy ($W_f$ > 1) can also reduce surfzone exits by increasing shoreward Stokes drift and broken wave bores (Castelle *et al.*, 2016) within the river channels. In the context of river/estuary discharges, this surfzone retention has been shown to be controlled by the ratio of ratio of river momentum flux to wave momentum flux (Olabarrieta *et al.*, 2014; Rodriguez *et al.*, 2018) or similarly of river plume length to surfzone width (Kastner *et al.*, 2019). In the context of bathing hazard, this is akin to breaker saturation reducing rip current exits through an inner bar-rip channel under larger waves

(MacMahan *et al.*, 2010) or above average wave power (Scott *et al.*, 2014). At Crantock, this effect doesn't seem to occur

during the lower tidal stages, however, with exits remaining at 15–20% for $1 < W_f < 4$ due to the well-defined boundary rip channels extending beyond the surfzone.

We present $U_{off}$ and $E$ in terms of relative wave power as Scott *et al.* (2014) and Castelle *et al.* (2019) found this to be an important parameter in controlling the occurrence of rip-related bathing incidents in southwest England and southwest France, respectively. Although they studied only a limited range of wave periods, Moulton *et al.* (2017a) and Moulton *et al.* (2017b) did not observe a dependence of rip current velocity on wave period and concluded that only wave height and direction (as well as water depth) were important for offshore directed flow velocity, due to their control on breaker-induced setup and alongshore current speed. Here we find that surfzone exits are slightly more sensitive to relative wave power (incorporating wave period) than wave height alone. Below mid tide, when the estuary is inactive, $U_{off}$ and $E$ varied up to 0.16 m/s and 51%, respectively, when averaged at each simulated wave height, while changing the level of wave power varied $U_{off}$ and $E$ by up to 0.17 m/s and 63%, on average. The simulations therefore indicate that seaward wave-driven velocity at an embayed beach is influenced to a similar degree by either wave height or power, but that wave power exerts a greater influence on surfzone exits than wave height alone.

### 6.3 River-channel bathymetric rips

During both the ebbing and flooding high tide, the estuary dictates the flow velocity in the river channels (Figure 10). Switching estuary discharge off in the model indicates that wave-driven rip currents in the deep river channels flow at a similar velocity to typical channel rips (~0.6 m/s) under average wave conditions (Section 4.3.2). These 'river-channel bathymetric rips' fit with the concept (McCarroll *et al.*, 2018) that intense rip flows occur in shore-normal channels with high alongshore non-uniformity (i.e., deep and narrow), regardless of whether the channels were formed by estuarine or wave processes. This also demonstrates that river channel morphology on a beach can facilitate both strong estuary flows and strong rip current flows, regardless of the level of estuarine discharge. Furthermore, the channels efficiently convey water towards wave-driven rip currents further down the beach, linking flows across the surfzone and providing a conveyor belt to transport bathers from the shore to deeper water offshore.

It is also noteworthy that the main river channel tends to exit seaward in approximately the middle of the embayment, albeit with some variation in its position (Section 4.3.3). Narrow embayments with curvature at the shoreline such as Crantock can promote cellular rip circulation (Castelle and Coco, 2012), where seaward flows form in the centre of the bay, especially during energetic conditions (Castelle *et al.*, 2016). This wave-driven process may, therefore, influence the position of the river channel at Crantock, by enhancing seaward flows and promoting channelisation in the middle of the beach.

### 6.4 Importance of tidal level and phase

Channel rips have been observed globally to increase in intensity at low tide as a result of flow constriction through low-tide bar-rip channels and/or increased intensity of wave breaking over sandbars (Aagaard *et al.*, 1997; Brander and Short, 2001; MacMahan *et al.*, 2005; MacMahan *et al.*, 2006; Castelle *et al.*, 2016). As a result, low tide levels have been linked with higher occurrence of bathing incidents (Scott *et al.*, 2014; Castelle *et al.*, 2019; Koon *et al.*, 2023) and drowning risk (Koon *et al.*, 2023). In contrast, here we find a mechanism for dangerous seaward flows to occur during the high tide phase, either from

estuary discharge or as river-channel bathymetric rips. While the efficacy of these high-tide rips was sensitive to the estuary discharge, they were present whether the estuary was ebbing or flooding (for example, Figure 9). Koon *et al.* (2023) found that on Australian beaches, there was no link between tidal phase or tide range, and coastal drownings. However, here we demonstrate a clear difference in bathing hazard during rising and falling stages of the tide when an estuary mouth is involved, albeit in a location with a much larger tidal excursion. During the upper half of the tidal cycle, hazard is strongly controlled by

tidal phase, range, water level, and wave conditions, while during the lower half of the tidal cycle only water level and wave conditions appear to be important.

**6.5 Bathing hazard forecasts**

Rip current forecasts have been developed in several previous studies. Some of these systems operate at a regional/national scale using data-driven empirical relationships that link forecasted wave, tide, and weather conditions to lifeguard rescue

statistics (Lushine, 1991; Lascody, 1998; Engle *et al.*, 2002; Dusek and Seim, 2013a, b; Gibbs *et al.*, 2015; Scott *et al.*, 2022). Other studies have developed site-specific calibrated process-based models to predict where and when rip current activity will occur on a beach (Austin *et al.*, 2013; Kim *et al.*, 2013; McCarroll *et al.*, 2015; McCarroll *et al.*, 2018), but these have rarely been applied operationally and none have yet included dynamics from channel rips, boundary rips, and estuary flow. The forecast system developed for Crantock (Section 5) has been implemented operationally at the beach since 2022 and provides

real-time warnings about where and when peak bathing hazards will occur, in addition to simplified flow visualisations, via novel digital display screens located at the two main beach access points (Figure 14). To the best of our knowledge, this represents the first process-based forecast system used to provide bathing warnings directly to the public. Work is now underway to better understand whether such warning systems are effective at influencing and informing beach user decision-making, and therefore contribute to a reduction in life risk.


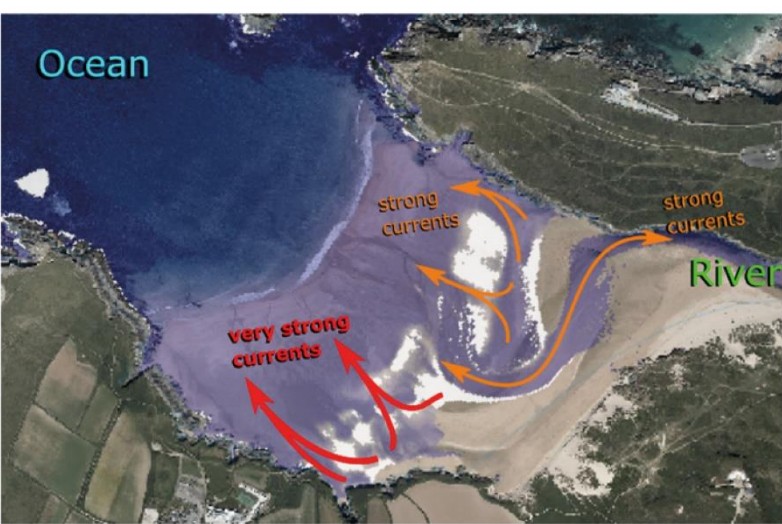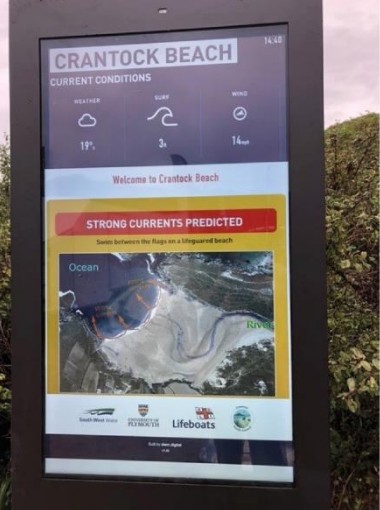

**Figure 14: (Left panel) example of simplified beach hazards information derived from simulated drifter patterns. (Right panel) operational digital bathing hazards sign at Crantock Beach next to a key beach access point showing dynamic hazard predictions for the next hour. Aerial imagery courtesy of National Network of Regional Coastal Monitoring Programmes of England, © 2024 NNRCMP.**

## 6.6 Limitations

The developed XBeach model, and the forecast system derived from it, show skill in capturing the circulation patterns and associated bathing hazards at this embayed, estuary mouth beach. However, there are several limitations to acknowledge that could be tackled in future research at similar sites:

- The simplified estuary discharges applied at the boundary ignore any potential gradients in water level over the length of the estuary and ignore some fluvial-estuarine interaction. However, enhancement of the estuary flow due to high fluvial flow was included by adding a conservative 5% exceedance river flow.

- While the observed circulation behaviour was overall reproduced by the XBeach model and the NRMSE in flow velocity was comparable to other similar studies (for example, Mouragues *et al.* (2021)), the modelled velocities were at times underestimated by the developed model (Section 4.2). Model bias was -0.07 m/s for flow magnitude, compared to the ADCP measurements. Therefore, the modelled flows do not always represent a conservative estimate of the real flow speeds.

- The surfbeat mode of XBeach was employed in this study, which captures the wave variations and associated wave-driven flows at the wave group (infragravity) timescale (Roelvink *et al.*, 2010) expected to drive the bathymetric and topographically controlled rips at Crantock (Austin *et al.*, 2010; Austin *et al.*, 2014; Scott *et al.*, 2014). However, transient flows driven at the incident wave timescale such as flash rips (Castelle *et al.*, 2016) are not captured by the model, which may occur over the planar lower beach morphology (Castelle *et al.*, 2014) away from the headlands.

- The influences of wave directional spreading and bimodality in the wave spectra have not been explored in this paper.

- The model is depth averaged, meaning that vertical stratification in flow is not considered. However, this is not considered to be important for studying hazard characteristics as only surface flows are of interest.

- The results of the present study are highly tuned to this specific estuary. For example, the flow velocities and exit rates are likely to be a product of local factors such as geomorphic setting, estuarine tidal prism, and wave exposure, while the variation in circulation between low and high tide may be a direct result of the large tidal range. The ebb-dominance of the estuary and required discharge tuning coefficient $\alpha$ are also likely to be site-specific and driven by geomorphological factors. The embaymentisation and oblique angle of the beach to the prevailing wave approach is also expected to elicit specific flow behaviours (shadow rips, for example) that won't necessarily occur in the same way at other estuary mouth beaches.

- The high variability in the river channel morphology appears to not fundamentally vary the bathing hazard in terms of $U_{off}$ and $E$, based on the four bathymetric data sets that were collected (Section 4.3.3). However, a more dramatic

change in the river channel morphology could feasibly occur (For example, if the river channel were to be naturally or artificially relocated once again against the north headland), and this has not been simulated in the present study.

- Several other relatively predictable factors cause a bathing hazard at this and other similar beaches which are not considered by the developed forewarning system but could feasibly be included in future developments. For example, beach users frequently get cut off from the shore on the ebb-tidal sandbars during the rising tide and are forced to
enter the water unprepared. Plunging breakers at the shore ('shorebreak' conditions) can occur during low and high tide periods at Crantock when waves break on the steepest parts of the shoreface, and these can lead to surfzone injuries (Castelle *et al.*, 2019).


# 7 Conclusions

Surfzone currents at an embayed estuary mouth beach were both measured and modelled, revealing complex surfzone circulation patterns, including circulating, alongshore, and exiting flow regimes. The river channel morphology is a key driver of the circulation above mid-tide. The river channels act to constrain both estuarine and wave-driven currents, directing the
flows alongshore and offshore, often connecting with boundary and channel rip currents lower on the beach face. Flow velocities through the river channels were enhanced by increasing estuary discharge, increasing wave power, and decreasing water depth. Wave direction was also found to alter bathing hazards, hindering seaward estuary flows during shore-normal waves and exacerbating shadow boundary rips during obliquely arriving waves. Overall, tidal stage exerted the greatest control on surfzone exits and seaward flows at this embayed, estuary mouth beach.

The most hazardous flows are predicted to occur during ebbing high spring tides with wave power at or below average, when estuary discharge and wave driven return flows can flow seaward through the river channels unhindered by the surfzone. Under such conditions, the highest seaward velocities (up to 1.5 m/s) and maximum potential for surfzone exits (> 60%) occurred. While wave-driven channel rips have been widely observed to occur preferentially over low tide bar-rip morphology, we demonstrate a novel mechanism for 'river-channel bathymetric rips' to occur near high tide due to wave breaking over an ebb-
shoal delta, which can drive strong seaward return flows in the adjacent river channels, even in the absence of estuary discharge. The combined action of estuary and wave-driven flows on this beach generates seaward currents that are up to 50% faster than peak rip current velocities observed in the literature and are combined with very high surfzone exit rates. This indicates that the presence of an estuary mouth within an energetic surfzone poses a highly hazardous situation for bathers which was previously unstudied in the literature, despite potentially exceeding the hazard that would be expected from rip currents alone.
Surprisingly, despite significant spatio-temporal variability in the position of the river channels on the beach face, it was found to be possible to predict the timing and severity of past bathing incidents from model simulations, providing a means to simulate and forecast bathing hazards to forewarn bathers about times of peak bathing hazard.

## Code/Data availability

Code and data may be partially available upon request. RNLI lifeguard data are not available, as permission is required from
local authorities to use these data.

## Author contribution

CS, TP, GM, TS, and SI conceived of the presented idea. CS, TP, GM, and TS developed the theory. CS performed the computations and model simulations. TP and TS orchestrated and processed the field data collection. GM and TS supervised
the findings of this work. All authors discussed the results and contributed to the final manuscript.

**Competing interests**

The contact author has declared that none of the authors has any competing interests.

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
