# Peer review of "New insights into combined surfzone and estuarine bathing hazards"

_EGUsphere, 2024_

## Author Comment (AC1)

Thank you for your comments, these have greatly improved the paper.

**Hydrodynamics: The paper describes a range of simulations with varying tidal levels and wave conditions, primarily representative of relatively quiescent conditions (Hs < 1.4 m). However, only results for varying wave power and tide are presented here. I have two concerns regarding this: First, the time series shows several larger wave events and more oblique waves than represented in the selection of simulations. Why did the authors select the set of hydrodynamics to simulate? Are they representative of times when swimmers are at the beach? Considering there are many users in HS3 conditions, should larger wave events be considered? Second, wave power is the only wave property presented here (Hs * Tp). Wave direction is only mentioned as 'unimportant' during mid-tide but is not otherwise described in the paper. Presumably, wave direction is important if waves are adequately oblique to reduce the energy entering the embayment. Wave direction is also shown to be important for channel rip hazards (e.g., Dusek and Seim, 2013). Is there little dependence on wave direction here due to the embayment geometry? Additionally, previous work on channel rip currents (e.g., Moulton et al., 2017) suggests that wave height is important for rip speeds but does not incorporate wave period. Assuming wave breaking is triggered at a wave height to depth ratio, wave height defines surfzone width. By presenting these values solely as wave power (which combines both period and height), this paper potentially glosses over some of the dynamics relevant for 'blocking exits', etc**

The conditions we simulated cover a wider range of conditions than you mention in your comment - in Section 3.5 we had summarised the boundary forcing conditions as 'wave heights of 0.5–3 m, wave periods of 6–12 s, and wave approaches from 269°–304°'. The conditions therefore cover the full range of summer wave conditions experienced at the site and do not represent quiescent conditions as mentioned. However, to clarify this I've now removed these words and instead added a summary table of the forcing conditions, and amended the text in Section 3.5 as follows:

"Once the model was calibrated (Section 3.7), seventy-two combinations of wave and tide conditions were run in the model covering the full range of summer wave conditions (Table 1), with each set of wave conditions run over a mean neap tidal cycle and a mean spring tidal cycle (with 30 minutes spin up time). The most energetic conditions are approximately 3.5 times higher than the summer (June, July, August) average wave power, equivalent to approximately the 1-in-1 year return period and would be conditions under which the lifeguards would close the beach to bathers. Each 12-hour simulation was then divided into 1-hour tidal segments at 30-minute increments, providing 1,728 unique combinations of wave and tide forcing from which to evaluate circulation patterns and bathing hazard from the simulated flow fields."

Thanks for your suggestion to delve into the influence of wave direction in more detail. Wave direction was varied in the simulations, but only the mean wave direction was presented in the bubble plot in Figure 10. This was to simplify the results and to make the bubbles legible (i.e. not overcrowding the figure with too many data points). I have now added an additional few sentences (see below) to describe the effects of wave direction, in comparison to wave power and tidal stage. Following your suggestion of using wave height instead of wave power to present the results, I replotted figure 10 to see what the hazard pattern looks like when plotted as wave height (x axis) vs tide level (y axis). This doesn't change the overall conclusions one would draw from figure 10. However, conditions with the same wave height but different periods plot on top of one another,

making the figure harder to interpret than the original. While I take your point about wave height being a key parameter controlling exits, and acknowledge that Moulton observes this, the Moulton study uses a limited range of wave periods (5-10 s observed, 7 s modelled). Furthermore, from the new analysis mentioned below, surfzone exits at Crantock are slightly more sensitive to wave power than wave height. Therefore, following your suggestion, I've kept the summary plot in Figure 10 the same, but have added the following text to comment on the influence of wave height, period, and direction:

"Figure 10 is presented in terms of relative wave power as Scott et al. (2014) and Castelle et al. (2019) both found this to be an important parameter in controlling the occurrence of rip incidents in southwest England and southwest France. Although they studied only a limited range of wave periods, Moulton et al. (2017) did not observe a dependence of rip current velocity on wave period and concluded that only wave height and direction (as well as water depth) were important for offshore directed flow velocity, due to their control on breaker-induced setup and alongshore current speed. Here we find that surfzone exits are slightly more sensitive to relative wave power (incorporating wave period) than wave height alone. Below mid tide when the estuary is inactive, $U_{off}$ and E varied up to 0.16 m/s and 51%, respectively, when averaged at each simulated wave height, while changing the level of wave power varied $U_{off}$ and E by up to 0.17 m/s and 63%, on average. The simulations therefore indicate that seaward flow velocity is influenced to a similar degree by either wave height or power, but that wave power exerts a greater influence on surfzone exits than wave height alone. Wave direction also appears to play some role in controlling the exit potential at Crantock. Below mid tide, wave direction varied $U_{off}$ by only 0.008 m/s on average, but E increased by 12% when wave direction was varied from the most oblique wave approach simulated (45°) to a shore-normal wave approach (0°). Wave directional influence is likely to be limited, however, by the highly embayed nature of the beach. Overall, the simulations indicate that tidal stage plays the most important role in controlling both surfzone exits and the velocity of surfzone currents at an embayed, estuarine beach like Crantock, as it varied $U_{off}$ and E by up to 0.44 m/s and 70%, respectively, when averaged at each tidal level."

**Hazard quantities: The paper primarily relies on two metrics to define hazard: maximum offshore flows and percent surfzone exits. Both are helpful metrics to assess bather hazards. Percent surfzone exits represent a free-floating bather's likelihood of being ejected offshore (but does not represent speed). Maximum offshore flows target how fast a bather may be advected offshore and, therefore, the feasibility for a bather to react (swim) from an offshore flow (but does not represent the offshore flow distribution). These metrics do not represent the number of locations with sufficiently strong offshore flows to eject a swimmer offshore. Providing an additional metric could help. For example, this could be represented as the percent of the locations alongshore with $U_{off}$ exceeding a threshold value or, possibly, the $U_{avg}$ and the rms alongshore of $U_{off}$. This similarly ties into the section on how these flows change with morphology. While $U_{off,max}$ does not vary strongly, the position and possibly this distribution of these flows change. This could be explored with an additional hazard metric.**

While I appreciate the point you are making about using another hazard metric that describes the spatial distribution of offshore flows, at this beach feeder currents are ubiquitous and direct bathers towards offshore flows that are either situated in the river channel, or in the headland rip channels at either side. We did test uoff over different sections of the beach (south and north) and in fact use

these in the live forecast system, but including this in the paper over complicated the results, and didn't alter the key conclusions. Furthermore, the spatial distribution of the offshore flows is described (at least qualitatively) by the spatial plots in figures 9, 11, and 13, and in the text. Many previous studies use only the offshore directed flow velocity (for example Moulton et al. (2017)), and therefore it seems unnecessary for this study to use more than two hazard metrics, although I can appreciate the motivation for doing so that you suggest. The two hazard metrics used adequately capture the occurrence of past incidents (Section 5), as was also found in a previous study (Austen et al., 2013). However, we acknowledge that spatial distribution is not well captured by the metrics we used in the paper, so I have added the following to the text at the end of Section 3.6:

"To forecast bathing hazard (Section 5), Uoff and E were quantified at each time step across three different sections of the beach (northern half, southern half, and estuary mouth) to acknowledge the fact that offshore flow velocity varies in different places along the shore and to differentiate the hazard a bather might experience in one part of the beach from another. However, given the large range of forcing and bathymetric combinations described in Section 4, the results presented in Section 4 summarise the variables as a single value across the beach for brevity."

**Hazard forecasts: While most methods and results are thorough, the forecasting bathing hazard section needs to be better described. I found the definitions of different terms and how the hazards were predicted and allocated difficult to follow, especially since some definitions are different in the figure caption. For example, the use of seemingly redundant terms (risk, hazard) and the overly brief explanation of the hazard scoring.**

Thanks for your feedback on this section; on reflection I agree that this section was far too brief and not clear. I have now re-written many of the paragraphs in this section in order to remove the redundant terms (e.g. risk is no longer referred to) and given a more detailed explanation of the hazard scoring.

**Limitations: The paper should describe the limitations of these hazard predictions. For example, the modeled velocities are often underestimated, resulting in non-conservative hazard estimates. The surf-beat depth-averaged model cannot represent all rip current types (e.g., flash rips). The findings are highly tuned to this specific estuary and may not represent other combined surfzone and estuary flows.**

We have now added a limitations section following your suggestion. This outlines the limitations you suggest, and also those around other relevant bathing hazards not considered by the forecast system.

**Figure quality: Figures 9 and similar layouts are very challenging to read. The quivers are barely visible, and the figures are grainy. Consider using plots similar to Figure 8c,f.**

Thank you for this suggestion. On reflection, I fully agree that those figures are too busy and that coloured quivers are not the best mechanism to show the flow behaviour. I have now replaced Figures 9, 11, 13 with simplified figures showing only the wave dissipation, depth contours, and bin averaged lagrangian velocity as colours (not quivers).

**Line-by-line:**

**L126: I expect the SfM DEMS to perform well at GCP locations because those locations were input into the algorithm to resolve camera geometry. Thus, this may not represent the accuracy of the DEM well. Can the DEM accuracy be checked by comparing the regions overlapping with both surveys (intertidal zone)?**

The SfM DEMs were not developed using the GCPs to tie in the DEM. The drone used has RTK capability, so positioning was provided by the on-board RTK corrections. The GCPs were only used to check the accuracy of the derived DEM and are therefore considered a suitable ground-truth of the DEM data. I have now added the following text to clarify this point:

"The DEM achieves a vertical RMSE of 0.03 m compared to independent spot checks against ground control points not used to geolocate the DEM during processing."

**L133: How was this RMSE computed? By comparing with?**

Thanks for pointing out this omission. I've now added some text to clarify:

"The bathymetric survey achieves a vertical RMSE of 0.1 m in the intertidal region, when compared to the previously mentioned ground control points"

**L143: List drifter dimensions. Is it truly a surface drifter or representative of a depth-avg current?**

The drifters are submerged approximately 0.5 m deep, so they represent a surface current in all but the shallowest areas. I have now added text to clarify this:

"The drifters were submerged approximately 0.5 m beneath the surface and therefore mapped the surface flow patterns."

**Figure 5: Define acronyms in Figure.**

We have now added definitions for all acronyms in Figure 5.

**L264: How do these tuned values compare with previous studies?**

I have now added comparison of tuning parameters with a previous (comparable) rip modelling study.

**L336: Report bias since the sign is important here (i.e., if the flows are over or underpredicted).**

Thanks for this suggestion, which I agree is useful. I have now added bias as a metric.

**Figure 8: Why are there large drifter passes at single points in the inner surf zone? Do drifters stagnate there?**

This is an interesting observation. These are quiescent areas in the inner surfzone where low velocity circulation between onshore and offshore flows is occurring. Therefore, I would agree that drifters are stagnating There. I have added the following text to Section 4.2 to reflect this observation:

"Interestingly, large numbers of virtual drifters are predicted to pass through certain points in the inner surfzone (**Error! Reference source not found.**, panel e), which are interpreted as stagnation points where quiescent circulation between onshore and offshore flow is occurring. These features

are not present in the real GNSS drifter tracks because of the statistical limitations of deploying a small fleet of drifters, but the velocity patterns in these areas are reproduced."

**Section 4.4: Is this supposed to be a new section?**

Thanks for pointing this out. This has now been changed to a new sub-section (4.3.1… etc).

**Figure 10: Error in the x-label -> HsTp/overbar{HsTp}.**

Thankyou, this has now been corrected.

**Section 4.6: The introduction claimed that the increase in swimmer rescues in recent years may be due to changes in the river channel. Here, the authors could add a simulation with synthetic bathymetry representing the previous, potentially less hazardous river channel to see how it compares with these morphologies.**

Thankyou for this suggestion. While I agree this would be insightful in clarifying the contribution of the change in river channel to the increased hazard, it would require a significant modelling effort to generate a reliable synthetic bathymetry (in the absence of a measured bathymetry from that time) and re-run a comparable set of scenarios. Furthermore, we don't even know the depth of the river channel at that time, and this would elicit considerable uncertainty in the predicted velocities. Therefore, I have instead added a point to the new limitations section of the discussion that this would be a desirable future area of study:

"The high variability in the river channel morphology appears to not fundamentally vary the bathing hazard in terms of $U_{off}$ and $E$, based on the four bathymetric data sets that were collected (Section 4.3.3). However, a more systematic change in the river channel morphology could feasibly occur, and this has not been simulated in the present study. For example, if the river channel were to be naturally or artificially relocated against the north headland (Section 2), then $U_{off}$ and $E$ near the estuary could be significantly increased as the channel would likely be straighter, narrower, and deeper. Conversely, $U_{off}$ and $E$ on the remainder of the beach could be drastically reduced, as the control of the river channels and ebb-shoal delta on the flows would be lost, as was the case prior to 2015. Lifeguards believe that it is the increased spatial variability in the flows, and resulting increase in bather exposure, that has increased the lifesaving burden on the beach. Future iterations of this research may seek to verify the additional hazard posed by the new river position compared to its former position against the north headland, by performing simulations with a synthetic river channel morphology that mimics the former river position."

**L555: The spatial variability described here should be incorporated more often throughout the paper.**

Thanks for this suggestion. I have now added text in the results section to mention the spatial variability.

**L600: NOAA's rip current hazard forecasts consider hydrodynamic conditions (https://oceanservice.noaa.gov/news/apr21/rip-current-forecast.html). Perhaps specify that this is the only forecast model considering estuarine flow.**

We appreciate that NOAA's rip current forecast does consider hydrodynamic conditions in terms of forcing. What was meant by this sentence is that this is the first forecast that provides detailed hydrodynamic predictions (in terms of temporal and spatial variability in the flow field), which NOAA's forecast and others does not (i.e. they simply provide rip warning levels). I have now clarified this sentence to make this point clearer to the reader, as we are not trying to detract from the excellent information that those systems provide at all:

"The forecast system developed for Crantock (Section 5) has been implemented operationally at the beach since 2022 and provides real-time warnings about where and when peak bathing hazards will occur, in addition to simplified flow visualisations, via novel digital display screens located at the two main beach access points (**Error! Reference source not found.**). To the best of our knowledge, this represents the first process-based forecast system used to provide bathing warnings directly to the public."

**Reference:**

**Dusek, G., and H. Seim. "Rip current intensity estimates from lifeguard observations." Journal of Coastal Research 29.3 (2013): 505-518.**

Thank you, I have now added this citation.

**Moulton, Melissa, et al. "Comparison of rip current hazard likelihood forecasts with observed rip current speeds." Weather and Forecasting 32.4 (2017): 1659-1666.**

Thank you, also now incorporated.

---

## Author Comment (AC2)

**General comments:**

**This observational and modeling study explored swimmer hazards in an understudied setting, where estuary mouth flows encounter surfzone currents including bathymetric rip currents and boundary rip currents with large tidal variations. The authors found that river channel morphology can facilitate not only strong estuary flows, but also strong rip current flows when the river channel modifies wave breaking patterns, similar to what occurs in a surfzone bar-channel system. In addition, prior studies of surfzone hazards have typically found hazard to depend on the water level, which modifies wave breaking, with no dependence on the tidal phase (ebbing vs flooding); in contrast, this study found bathing hazard was different during rising versus falling tides when ebbing or flooding estuarine flows were interacting with the surf zone. I found these conclusions to be very interesting, novel, and supported by the analysis. I do have several concerns about (1) the framing of the paper, (2) the forecasting hazards analysis, and (3) the clarity of the text and figures.**

Thank you for your comments, these have greatly improved the paper. I am pleased you found the conclusions interesting, novel, and supported by the analysis.

**General comment (1): The paper emphasizes estuary mouth flows and bathymetric rip currents. Some discussion of headland/boundary rip currents is included but should be expanded given the clear importance of the boundary flows in this system. In addition, the paper lacks discussion of other rip-current types like flash rip currents, which I would expect to be present, as well as embayment rip currents, which would form in the center of an embayment rather than at the boundaries. Given the importance of the flows resulting from the embayment geometry, I wonder if the title and framing of the paper should be adjusted to "Combined surfzone, embayment, and estuarine bathing hazards."**

While we recognise that boundary flows are important at this site and discuss them in the results and to some extent in the discussion (including comparing velocities and surfzone exit rates to those found in other boundary rip current studies), the paper's focus and novelty is on what happens when estuary flows enter a surfzone. To maintain a reasonable word count, boundary rips and flash rips are not dealt with in detail in this paper. However, as boundary rips are mentioned in the results, I've added the following text to the introduction:

"The body of previous research has demonstrated various forcing mechanisms for rip currents (Castelle *et al.*, 2016), including hydrodynamic instabilities in the surfzone ('shear instability rips' and 'flash rips'), bathymetric control of wave breaking and return flows ('channel rips' and 'focus rips'), and control of wave driven flows by headlands or other boundaries ('deflection rips' and 'shadow rips')."

And have added the following text to the discussion about cellular rip currents:

"It is also noteworthy that the main river channel tends to exit seaward in approximately the middle of the embayment, albeit with some variation in its position (Section 4.3.3). Narrow embayments with curvature at the shoreline such as Crantock have previously been demonstrated to promote cellular rip circulation (Castelle and Coco, 2012), where seaward flows form in the centre of the bay, especially during energetic conditions (Castelle *et al.*, 2016). This wave-driven process may, therefore, influence the position of the river channel at Crantock, by promoting channelisation in the middle of the beach and enhancing seaward flows."

I've also mentioned flash rips in the new limitations section of the paper:

"The surfbeat mode of XBeach was employed in this study, which captures the wave variations and associated wave-driven flows at the wave group (infragravity) timescale (Roelvink *et al.*, 2010). Therefore, transient flows driven at the incident wave timescale such as flash rips (Castelle *et al.*, 2016) are not captured by the model. However, given the topographic control over wave breaking and circulation on this coastline (Austin *et al.*, 2010; Austin *et al.*, 2014; Scott *et al.*, 2014), bathymetric and topographically controlled rips driven by wave group scale forcing are far more common."

**General comment (2): The forecasting bathing hazard section isn't clearly described and is lacking important details. In particular, the analysis relies on a look-up table of hazard statistics from a prior study, but the authors don't provide a summary of this study or it's applicability to their study. The authors emphasize how this study site and combination of processes is understudied, so it warrants some explanation why a hazard model developed for a different setting would be the right choice here. The relationship between risk, hazard, and exposure isn't explicitly stated, and there is some redundancy in presenting all of these separately. Proportions of hazard scores are presented, but a forecast skill assessment should be performed.**

Thanks for your feedback on this section; on reflection I agree that this section was far too brief and not clear. I have now re-written many of the paragraphs in this section in order to remove the redundant terms (e.g. risk is no longer referred to) and given a more detailed explanation of the hazard scoring. Importantly, I've clarified how we optimised the hazard scoring based on our data, as the thresholds were not from the previous paper, just the overall approach of combining seperate hazard scores. Traditional skill scores cannot be applied as the predicted variables ($U_{off}$ and E) don't directly match any of the measured variables (incidents, hazard, exposure), so we can't compute R2, RMSE, AIC, etc. Instead, I've clarified how we used the Probability Of Detection metric (also known as Recall or Sensitivity), which is simply the rate of true positives and false negatives achieved, as this is the only useful skill score that can be determined from this sort of data. I've now added significant text to Section 5 to explain this.

**General comment (3): Prior to publication I think some improvements to the clarity of the text and figures are needed, particularly to emphasize limitations of the study and make sure that key results are discernable in the figures (see Specific comments).**

Thank you for this valid suggestion. I have now included a limitations section within the discussion and have improved the clarity of Figures 9, 11, and 13, by using a similar style to Figure 8c/f.

**Specific comments:**

**L38-43: This text suggests that rip-current patterns/behaviors, rather than speeds, classify the hazard level. This is then inconsistent with the next statement that places importance on speeds. I think this section would be clearer if the authors started with a statement that it is expected that a combination of factors, including pattern and speed, influence the hazard.**

I have now included the following text there:

"On beaches with rip currents, a combination of factors, including circulation pattern, speed, and surfzone retention influence the bathing hazard."

**L46: Here it is concluded that flows in estuary channels may pose "an equal or potential even higher bathing hazard than rip currents" because the speeds are equal or greater than the speed of rip currents. I don't think this is known.**

I'm not quite sure what you mean here – do you not think that the velocities are greater in an estuary, or do you not think that the hazard could be equal or higher than that of a rip current? Hopefully we have provided sufficient evidence from the literature to pose this as a hypothesis at least. We are not claiming to know this with certainty, just pointing out that the literature suggests that (a) estuary flows are strong relative to rips and (b) this represents an understudied hazard (as flow speed is routinely related to hazard level in rip studies).

**L75: "embaymentisation ratio (length/depth)" Maybe "headland amplitude" and "embayment width" would be clearer terms than depth/length? Labeling these scales in Figure 2 could be helpful too.**

I have added "(alongshore length/headland length)" to the text to clarify this.

**L75-77: When do boundary rip currents occur, in which fast flows are along the headlands, versus a headland circulation, in which the fast flows are in the middle of the embayment? If a headland circulation is occurring some of the time, could this enhance the flows out of the estuary channel?**

This is a good point, and rather than mentioning it briefly in Section 2, I've added it to the discussion section, as per General Comment (1) above.

**L82: It may be helpful to label features such as "ebb tide delta" on the figure.**

Thanks for this suggestion, I've added that to the figure.

**L98: Did you introduce available data on water users?**

This is now mentioned in Section 5 where we talk about the lifeguard head counts

**L133: Give some information on how the echosounder and UAV datasets are merged. Is this the same as the process described later for the model bathymetry?**

I've now signposted section 3.4, which explains the merging of the survey data sets

**L136: Flow measurements were at 0.1 m above the seafloor. It would be worth describing why this is expected to be a good representation of swimmer hazard, and if there are times when it might not be.**

The Eulerian measurements are not used to describe swimmer hazard. We are primarily using them to characterise surfzone velocities where estuary flows occur, and further using them for model calibration/validation.

**L144: Are there concerns about when drifters may be poorly tracking the currents, e.g., if they are scraping the seafloor or surfing waves?**

I've added the following text to clarify this:

"Lagrangian measurements were collected using GNSS-tracked surfzone drifters (**Error! Reference source not found.**), which are designed to mimic a static bather being carried by the surface flows (submerged approximately 0.5 m beneath the surface) and avoid surfing landward on waves. These were telemetered in real-time allowing shore based logging using QPS Qinsy software package (following Mouragues *et al.*, 2020). Six drifters were deployed at numerous locations multiple times across the survey area throughout the tidal cycle and were retrieved from the shallows before they ran aground."

**L175: can delete "respectively"**

Done.

**L200: "tidal variation was imposed uniformly on all four corners of the model domain" – what does this mean?**

This means that the water level was increased in a spatially uniform manner, i.e. there is no tidal gradient across the model domain. I've altered the text to clarify this:

"tidal variation was imposed uniformly across the modal domain."

**L206: Wave directions are mentioned here, but not wave directional spread, which seems to vary tidally in the observational record. How does this influence the results of this study? I don't think spread was varied in the model runs? In addition, it would be helpful to know if the observed wave spectra are well described by a single peak period and direction, or if a wave systems approach would be more accurate. How does this affect the results? I suggest referencing the observations here to say how the model spans the observational conditions (shown for this year, is this similar for other years?).**

Thanks for this suggestion. We haven't explored the influence of wave directional spreading in this paper, although it was varied during the calibration and validation runs. For the 72 model runs used to populate the hazard look up table we used the average spreading value for the site of 30 degrees. I have now mentioned spreading in the text and added a table of forcing conditions in Section 3.5 to clarify this. We also do not explore bi-modality in the spectra in this study, because it is not deemed important at this site due to the narrow range of wave approach angles on this coast and lack of energetic wind waves during typical bathing conditions. I have added some text to the new limitations section to reflect these omissions:

"The influences of wave directional spreading and bi-modality in the wave spectra have not been explored in this paper."

**L213-223: Does this method of forcing the estuary flows miss any river-estuary interactions, or does it include them because it's based on a measurement near the estuary mouth? A small discharge is added to "conservatively account for fluvial flow" – can you elaborate and say if the results are sensitive to these choices?**

The river input is small relative to the estuary input (typically <2% of the spring tide discharge) at this beach and is expected to also be small at other similar sites (as per reference in first para of introduction). The only river-estuary interaction I can think that would be relevant to bathing hazard is an enhancement of flows, which has been accounted for conservatively by adding the 5% exceedance river discharge (which rarely occurs during the summer bathing season). The surfzone flows are not sensitive to this level of discharge, as we found in initial tests. I have added the following text to clarify this:

"For the scenario simulations, the discharge applied at the boundary was computed from the estimated spring and neap tidal discharge rate at a given point in time, plus an additional 2 $m^3$/s to conservatively account for fluvial flow (5% exceedance river discharge). However, initial tests with only fluvial discharge applied showed that this fluvial discharge rate has a negligible effect on surfzone flows."

**L232: "Each virtual drifter was advected for 20 minutes, or until they had returned to a safe water depth (<0.7 m)." Could you elaborate on these choices and how they affect the results? Why 20 minutes? Why is a safe water depth 0.7 m? Why not keep running the drifters to see what happens next even if they enter safe water?**

0.7 m is a minimum 'safe' depth taken from the cited studies by McCarroll (2014 and 2015). I have added text to clarify this:

"Depths shallower than 0.7 m are deemed 'safe' as bathers can stand up without being swept off their feet by typical surfzone currents."

I have also added the following text to elaborate on the 20 minute timeframe:

"The 20 minute timeframe was chosen to represent a typical timescale of a bathing incident – it is likely that a person in a strong current would either be rescued or in a critical state within 20 minutes. Furthermore, as we simulate with non-stationary tides, leaving drifters to circulate for longer blurs the effects of different tidal stages."

**L246: should this be lowercase u_off?**

Thank you, now corrected.

**L328: Mention what may be the cause of these fluctuations (e.g., related to infragravity pulsations, instabilities, or flash rip currents?).**

We did not study the time-variation in the measured drifter velocities in this paper. The mentioned average velocity is a spatial average. I have added the following text to clarify this:

"The spatially-averaged lagrangian velocity during this phase of the tide was 0.3 m/s, with peak velocities exceeding 0.6 m/s…"

**L355: Given the spatial complexity of the flows, a point to point agreement may not be expected. You could plot the modeled maximum flows and flow range within a spatial region around the observational point for a more fair comparison?**

We completely agree, the stochastic nature of the wave-driven flows means we would never expect the real/virtual lagrangian driftrers to agree exactly. The purpose of the lagrangian comparison is to

check qualitatively that the spatial circulation pattern is approximated by the model, while the Eulerian data are used to provide quantitative comparison. This is explained in the text.

**L395-404: There is little or no discussion of rip currents here. How does the variation in rip current strength and characteristics with wave power or other factors influence the hazard metrics?**

Rip currents are not the focus of this paper, and we are not trying to characterise changes in rip flow under different forcing. The focus of the paper is to characterise how estuary flows influence bathing hazard when they enter a surfzone. Rip currents occur at this beach during mid and low tides, but we are mainly using the low tide E and Uoff values to provide context to the values driven by estuary flows at high tide. E.g. this beach has typical rip current characteristics at mid-low tide, but enhanced flows at high tide due to the estuary. This is discussed in detail in the discussion section

**L427-429: The ebb shoal delta acting as a bar rip system is interesting. Is this mentioned elsewhere in the literature, e.g., papers on flows near a small river mouth encountering a surf zone (Kastner et al., Rodriguez et al., 2018).**

Thanks for these references, I had previously struggled to find existing examples of this in the literature. From Kastner et al I also found a good citation (Olabarrieta et al) for wave-driven flows over an ebb-shoal delta. I have now included these in the introduction and discussion

**L510: It seems the authors are following a prior method (Austin et al., 2013), but more explanation here of how these thresholds and scores were developed is needed to understand this section.**

This has now been expanded and explained better. We are not using the thresholds of the previous paper

**L512: Assuming I'm correct that Risk = Hazard x Exposure (which isn't spelled out here), and Exposure is a measured quantity, it seems redundant to me to present accuracy results for both Risk and Hazard.**

This section has now been re-written to better explain these variables. To lifeguards who will ultimately implement the science developed here, understanding how many incidents occur at each predicted hazard level is of equal importance to understanding how much hazard occurs. Also, while assessing how the observed level of hazard increases at each skill score tells us qualitatively how the system performs as a hazard forecast, the number of incidents (risk) can be used to determine the predictive Recall (rate of true positives/false negatives), which is the most useful skill metric that can be generated from this data.

**Figure comments:**

**Figure 1: x and y axes are not labeled in the top panel.**

This has now been clarified in the caption

**Figure 5: The panel b transect where the Gannel estuary enters the beach shows the topography over a long distance, which does not seem necessary, and makes it harder to see the channel. It may be more useful to see a transect across the shoals to show the scales of the estuary channel and other channels connected to the swimmer hazards.**

This has now been zoomed in to show the channel morphology more closely

**Figure 6: Were other days similar? This could be interesting to look at as a composite, though maybe that would be difficult given the variation in the behavior with tidal range.**

Other days were very similar in the overall signature. I've added the following text to Section 4.1 to mention this:

"Each of the measured tidal cycles showed a similar hydrodynamic signature"

**Figure 7: Do other time periods look similar? It could be interesting to show a scatter plot comparing measured and modeled flows.**

Again, other tidal cycles were basically the same comparison, so we chose to show just two tidal cycles to demonstrate the pattern and ability to replicate the pattern. These and the skill scores are deemed sufficient to demonstrate the model fit to the data.

**Figure 8: This figure compares observed drift tracks with gridded results of model drift tracks. It may be helpful to also show example model drift tracks (not expected to reproduce the observed tracks, but presumably similar patterns), and gridded observational data, so that there's a more one to one comparison. Does this figure only show two of the three regimes described in the text (L374-379)? I suggest using colormaps that are colorblind friendly and perceptually uniform (e.g., Thyng et al., 2016).**

Thanks for this suggestion. I've now included some example virtual drifter tracks (showing all would cluter the figure). I've also changed the colormap to exclude the use of red and tested this on a colorblind simulator. The figure does shows the 'main' regimes of interest, i.e. the high-ebbing tide when the estuary is active and the low tide period when the estuary is inactive.

**Figures 9&11&13: Vectors are small and very difficult to see. Colormap for the vectors includes dark blue, which is the same color as the background water color. This figure shows the variation in the wave breaking, water levels, and inundated bathymetry, but is not readable for information about flow patterns.**

On reflection, I completely agree that this figure was not well conceieved. I've now simplified this significantly to show the seaward velocities and depth contours only.

**Figure 10: This is an interesting figure, though a bit difficult to interpret. It's clear there is a difference between rising tide and falling tide. Showing line plots of Uoff vs the wave factor for three example tidal elevations for rising and falling tides may be helpful as a summary of this figure.**

Thanks for this suggestion. I've now included an extra panel below the bubble plot to display this summary, as suggested.

**Figure 12: I'm not sure I find the change plots very helpful. It may be more useful to show how the flows differ for the different cases, to show different spatial patterns, but similar magnitudes?**

Figure 13 shows the mentioned flow comparison with different morphologies. On reflection, I agree that Figure 12 doesn't add much to the paper and in the interest of brevity I've decided to remove it.

**Figure 14: Is proportion the most useful comparison? This doesn't show anything about the timing of events. Why isn't a forecast skill assessment shown? I'm not quite sure what to take from this figure.**

Traditional skill scores cannot be applied as the predicted variables (Uoff and E) don't directly match any of the measured variables (incidents, hazard, exposure), so we can't compute R2, RMSE, AIC, etc. Instead, I've clarified how we used the Probability Of Detection metric (also known as Recall or Sensitivity), which is simply the rate of true positives and false negatives achieved, as this is the only useful skill score that can be determined from this sort of data. I've now added significant text to Section 5 to explain this.

**References:**

**Kastner, S. E., A. R. Horner-Devine, and J. M. Thomson (2019), A Conceptual Model of a River Plume in the Surf Zone, J. Geophys. Res. Ocean., 124(11), 8060–8078, doi:10.1029/2019JC015510.**

Thankyou, this is a highly relevant reference. I have now cited this in the intro and discussion

**Rodriguez, A. R., S. N. Giddings, and N. Kumar (2018), Impacts of Nearshore Wave-Current Interaction on Transport and Mixing of Small-Scale Buoyant Plumes, Geophys. Res. Lett., 45(16), 8379–8389, doi:10.1029/2018GL078328.**

Thankyou, this is a highly relevant reference. I have now cited this in the intro

**Thyng, K. M., Greene, C. A., Hetland, R. D., Zimmerle, H. M., & DiMarco, S. F. (2016). True colors of oceanography. Oceanography, 29(3), 10. link: http://tos.org/oceanography/assets/docs/29-3_thyng.pdf, https://matplotlib.org/cmocean/**

Thanks for this, we have now changed to a Matlab colormap (Parula) similar to their Haline colormap deemed suitable to colorblind readers

---

## Author Response (AR2)

**Replies to reviewer 1**

The authors thoroughly responded to reviewer comments. Thank you. I only have a few minor additional comments:

(1) The term 'river channel rips' is misleading. Rip currents are driven by the action of wave breaking (Bowen, 1969). In this case, the rips described are breaking patterns over bathymetric features formed by the estuarine processes. Thus, a more appropriate term would be 'river-channel bathymetric rip' or something along the lines of this rip current type being a sub-category of a bathymetric rip.

Thank you for this suggestion, I agree it is clearer to use that wording. I have amended this in the abstract, conclusions, and Section 6.3:

These 'river-channel bathymetric rips' fit with the concept (McCarroll *et al.*, 2018) that intense rip flows occur in shore-normal channels with high alongshore non-uniformity (i.e., deep and narrow), regardless of whether the channels were formed by estuarine or wave processes.

(2) The new findings suggest that mean direction is important. This should be reflected in the conclusion.

Good suggestion. I have now amended the first paragraph of the conclusions as follows:

'Surfzone currents at an embayed estuary mouth beach were both measured and modelled, revealing complex surfzone circulation patterns, including circulating, alongshore, and exiting flow regimes. The river channel morphology is a key driver of the circulation above mid-tide. The river channels act to constrain both estuarine and wave-driven currents, directing the flows alongshore and offshore, often connecting with boundary and channel rip currents lower on the beach face. Flow velocities through the river channels were enhanced by increasing estuary discharge, increasing wave power, and decreasing water depth. Wave direction was also found to alter bathing hazards, hindering seaward estuary flows during shore-normal waves and exacerbating shadow boundary rips during obliquely arriving waves. Overall, tidal stage exerted the greatest control on surfzone exits and seaward flows at this embayed, estuary mouth beach.'

(3) Perhaps I missed this, but how is an incident defined? A lifeguard rescue attempt?

This is defined on line 567 'Only flow-related incidents (n = 648) were considered where a lifeguard was required to rescue or assist a water-user back to shore (**Error! Reference source not found.**).'

(4) L571: How are the threshold values defined? Are these simply chosen as the best hazard agreement with observations by manual tuning? Please explain. Why are there only a low and high threshold for the exits?

I've now slightly re-worded this paragraph to make it clearer:

'The thresholds in Table 2 were optimised by analysing past bathing incidents at Crantock Beach over the years 2016–2021. Only flow-related incidents (n = 648) were considered where a lifeguard was required to rescue or assist a water-user back to shore (Figure 13). The lifeguard data were discretised into 2-hour time bins and the number of Incidents were divided by the bather head count made by the lifeguards during each 2-hour period (representing an estimate of the average Exposure over that period), resulting in an 'observed' Hazard level from Eq. 1 for each timestep. The Hazard timeseries was then used to compute bin-averaged Hazard values across a number of discrete $U_{off}$ and E bins. The distribution of Hazard over these bins suggests that sharp increases in Hazard occur

when Uoff reaches 0.2 m/s and 0.4 m/s. The lower threshold is corroborated by Moulton et al. (2017a), who identified that rip current speeds greater than 0.2 m/s may be hazardous to swimmers. For E we find a single threshold of 0.2 (20% likelihood of a drifter exiting the surfzone), which distinguishes between lower and higher levels of Hazard. An obvious second increase in Hazard with E was not visible from the distribution. Using these thresholds, two scores are obtained from Table 2 which are added together and rounded to achieve a final Hazard Score, following the approach of Austin et al. (2013).'

(5) Fig 13: The caption is not updated per the new section. For example, 'Risk' is still included.

Thankyou, this is now amended:

Performance summary of the developed bathing hazard forecast over the hindcast period (2016–2022). Proportion of forecasted Hazard Scores (HS1, HS2, HS3; upper left), relative average water-user exposure (upper right), proportion of total incidents (lower left), and probability of an individual water user being in a flow-related incident (Hazard, lower right).

(6) L718: Do we actually know that bathymetric and topographically controlled rips are far more common here? This study did not use a phase-resolved model, so the authors can't prove anything about transient rips with these findings. Is there clear evidence in the literature that transient rip processes would not be important here?

I agree, that was a stretch too far. I've now re-worded that bullet point to better reflect our understanding of their importance here:

'The surfbeat mode of XBeach was employed in this study, which captures the wave variations and associated wave-driven flows at the wave group (infragravity) timescale (Roelvink *et al.*, 2010) expected to drive the bathymetric and topographically controlled rips at Crantock (Austin *et al.*, 2010; Austin *et al.*, 2014; Scott *et al.*, 2014). However, transient flows driven at the incident wave timescale such as flash rips (Castelle *et al.*, 2016) are not captured by the model, which may occur over the planar lower beach morphology (Castelle *et al.*, 2014) away from the headlands.'

**Replies to reviewer 2**

This is the revised version of an interesting observational and modeling study exploring swimmer hazards in an understudied environment where estuary mouth flows encounter surfzone currents in an embayment. The authors addressed all of the minor line-by-line suggestions. My earlier concerns were (1) the framing of the paper, (2) the forecasting hazards analysis, and (3) the clarity of the text and figures.

For (1), I appreciate the mention of boundary rip currents in the intro and the added comment about cellular circulation is great. I still think the abstract (and possibly the title) should mention the embayed setting and boundary rip currents to accurately represent the paper and to highlight the unique setting that contributes to the paper's interest and novelty. This system has an estuary entering a beach within an embayment, which has modified wave conditions and boundary rip currents not present on an open-coast. The authors note this in the Discussion: "The embaymentisation […] elicit[s] specific flow behaviours (shadow rips, for example) that won't necessarily occur in the same way at other estuarine surf beaches." In addition to many mentions of

boundary rips in the results, the authors note in the hazard section that other forecasts have not "yet included dynamics from channel rips, boundary rips, and estuary flow," in the Discussion there is a section titled "Embayment, estuary, and wave controls on surfzone exits," and the Conclusions state that "The river channels act to constrain both estuarine and wave-driven currents, [...] often connecting with boundary and channel rip currents lower on the beach face." To me, the embayment setting and boundary rips are a significant part of the story of this paper, but if I'm misunderstanding, maybe the text needs to be revised some to reflect this.

On reflection, I agree that this has become (especially since the first round of revisions) a significant part of the paper. Therefore, I have amended the title as per your original suggestion to:

'New insights into combined surfzone, embayment, and estuarine bathing hazards'

And I've slightly revised the abstract to also reflect this:

'Rip currents are the single largest cause of beach safety incidents globally, but where an estuary mouth intersects a beach, additional flows are created that can exceed the speed of a typical rip current, significantly increasing the hazard level for bathers. However, there is a paucity of observations of surfzone currents at estuary mouth beaches, and our understanding and ability to predict how the bathing hazard varies under different wave and tide conditions is therefore limited. Using field observations and process-based XBeach modelling at an embayed, estuary mouth beach, we demonstrate how surfzone currents can be driven by combinations of estuary discharge and wave-driven bathymetric and boundary rip currents under various combinations of wave and tide forcing. While previous studies have demonstrated the high hazard that rip currents pose, typically during lower stages of the tide, here we demonstrate that an estuary mouth beach can exhibit flows reaching 1.5 m/s – up to 50% stronger than typical rip current flows – with a high proportion (>60%) of simulated bathers exiting the surfzone during the upper half of the tidal cycle. The three-dimensional ebb shoal delta was found to strongly control surfzone currents by (1) providing a conduit for estuary flows that connects to boundary headland rips, and (2) acting as a nearshore bar system to generate wave-driven 'river-channel bathymetric rips'. Despite significant spatio-temporal variability in the position of the river channels on the beach face, it was found to be possible to hindcast the timing and severity of past bathing incidents from model simulations, providing a means to forewarn bathers of hazardous flows.'

For (2), the revised and expanded forecasting section is much more detailed and clear.

Thankyou, your suggestions have definitely helped to improve this section.

For (3), the authors improved the clarity of the figures and discussion of the study limitations. I like the added time series in Figure 10 and the change to Figure 12 showing the signed cross-shore velocity in color is an improvement over the quivers. In Figure 8, the zoom regions could be shown with boxes in a, d.

Thankyou for this additional figure suggestion. I have now added zoom regions to in Figure 8, which I agree clarifies the zoomed panels.

I have no further comments.

Thankyou for taking the time to review our paper again.